Resource

# A molecular toolkit of cross-feeding strains for engineering synthetic yeast communities

Huadong Peng [1,2,3], Alexander P. S. Darlington[4], Eric J. South [5], Hao-Hong Chen[1,2,6], Wei Jiang [1,2,3] & Rodrigo Ledesma-Amaro [1,2] ✉

Engineered microbial consortia often have enhanced system performance and robustness compared with single-strain biomanufacturing production platforms. However, few tools are available for generating co-cultures of the model and key industrial host *Saccharomyces cerevisiae*. Here we engineer auxotrophic and overexpression yeast strains that can be used to create co-cultures through exchange of essential metabolites. Using these strains as modules, we engineered two- and three-member consortia using different cross-feeding architectures. Through a combination of ensemble modelling and experimentation, we explored how cellular (for example, metabolite production strength) and environmental (for example, initial population ratio, population density and extracellular supplementation) factors govern population dynamics in these systems. We tested the use of the toolkit in a division of labour biomanufacturing case study and show that it enables enhanced and tuneable antioxidant resveratrol production. We expect this toolkit to become a useful resource for a variety of applications in synthetic ecology and biomanufacturing.

Microbial communities have attracted interest due to their wide applications in industrial processes (such as the production of biochemicals[1], biofuels[2], biomedicines[3,4] and biomaterials[5]) and their important role in human, animal and crop health[6–8]. The composition and stability of these systems are influenced by various factors, including the chemical and physical characteristics of the environment, and the interactions between neighbouring microorganisms of the community[9,10]. Despite the importance of microbial communities, we still know little about how communities are established and maintained, which restricts our ability to engineer them for either improving human health or industrial purposes[7]. To this end, there is notable interest in developing simplified synthetic microbial communities, or consortia, that can both address basic biological questions on microbial interactions and create more efficient bioprocesses than those based on a single engineered microorganism[11,12].

Microbial interactions include commensalism, amensalism, neutralism, mutualism, competition and parasitism[13]. Syntrophy, otherwise known as obligate mutualism, is a cooperation strategy where microorganisms survive by feeding on the metabolic (by-)products of neighbours. Such metabolic co-interdependencies (that is, cross-feeding behaviours) are ubiquitous in natural communities[6,7,14]. In a consortium of co-auxotrophic strains, the survival of each member is dependent on other members supplying a particular nutrient which the recipient itself cannot synthesize. These nutrients could be amino acids, nucleotides or other essential metabolites[15–17]. For example, a two-member *Corynebacterium glutamicum* co-culture was created consisting of ʟ-leucine and ʟ-arginine auxotrophs[18], and various *Escherichia coli* co-cultures have been created in vivo or designed in silico, which have ranged from 2–14 auxotrophs[15,19,20]. Syntrophy promotes system robustness by preventing competitive exclusion

[1]Department of Bioengineering, Imperial College London, London, UK. [2]Imperial College Centre for Synthetic Biology, Imperial College London, London, UK. [3]The Novo Nordisk Foundation Center for Biosustainability, Technical University of Denmark, Kongens Lyngby, Denmark. [4]Warwick Integrative Synthetic Biology Centre, School of Engineering, University of Warwick, Coventry, UK. [5]Molecular Biology, Cell Biology and Biochemistry Program, Boston University, Boston, MA, USA. [6]School of Food Science and Engineering, South China University of Technology, Guangzhou, China. ✉e-mail: r.ledesma-amaro@imperial.ac.uk

between neighbouring strains and instead passively regulates community growth dynamics over time on the basis of nutrient availability[21,22].

Progress on establishing cross-feeding *E. coli* communities has been made[15,19,23], but engineering yeast communities is less developed despite yeast's wide use as a eukaryotic model organism and important industrial host. So far, there are only a few examples of how distinct combinations of strains elicit stable syntrophic phenotypes. These include non-mating *Saccharomyces cerevisiae* lysine-adenine or leucine-tryptophan auxotrophic pairs[24,25] and SeMeCo, a self-establishing, metabolically cooperating yeast community developed by randomly introducing auxotrophs into a population via loss of plasmids that express genes involved in amino acid and nucleotide biosynthesis[26,27].

Despite the many examples of microbial communities cooperating on bioproduction tasks, the relationship between population composition, growth dynamics and product formation remains undercharacterized. Microorganisms continuously respond to environmental cues, resulting in fluctuating growth rates that ultimately determine the composition and productivity of a community. Maintaining the stability of engineered communities remains a challenge, with various strategies proposed to control subpopulations and mitigate community collapse (for example, deep reinforcement learning[28], dynamic light inputs[29], use of multiple growth substrates[30,31], transcription factor-based biosensors[3], quorum sensing[23,32] or physical encapsulation[3]). These methods represent top-down approaches that attempt to stabilize microbial communities through artificial means. Bottom-up approaches, where stable communities are achieved from first principles by combining appropriately suited cross-feeding strains and environmental conditions, remain largely unexplored and yet may facilitate more robust, predictable systems in industrial settings. To predictably control the behaviour of constituent members in a microbial community, more experimental synthetic biology tools are needed. Efforts to design these systems are benefitting from static genome-scale metabolic modelling approaches, which are increasingly being used to both understand cross-feeding relationships of natural consortia and design syntrophic communities[20,33,34].

Here we present an ensemble dynamic modelling approach to identify the key factors that influence microbial community dynamics, and then establish a toolkit for engineering synthetic *S. cerevisiae* communities. Identifying metabolite exchange as a key factor that informs growth dynamics, we created 15 auxotrophic strains by engineering amino acids or nucleotide gene knockouts. We then built upon these auxotrophs to create overproduction strains for different intermediate metabolites. These strains represent modules that can be defined as 'donor' and 'receiver' cells in synthetic cross-feeding relationships. We demonstrate the use of our toolkit by establishing novel two- and three-member yeast co-cultures. Through ensemble modelling and experimental approaches, we demonstrated how different strategies, including metabolite production rate, metabolite supplementation, initial population ratio and initial cell density, can control co-culture dynamics. We used our toolkit to increase production of the high-value aromatic resveratrol by dividing its metabolic pathways between two strains. The presented toolkit has wide applications for both studying novel microbial communities and improving bioproduction of high-value compounds.

## Results

### Identifying key engineering targets in co-culture dynamics

There are numerous experimental interventions, referred to here as 'dials', available to manipulate the dynamics of microbial co-cultures. These include initial population ratio, different strain growth rates, culture supplementation and metabolite exchange. To explore these strategies, we first developed a nonlinear coupled ordinary differential equation model based on that previously proposed[20]. The model captures the time evolution of a microbial population and environment (made up of glucose and exchanged metabolites). Each strain $i$ takes up glucose ($G$) and its auxotrophic 'received' metabolite ($j$) at rates $J_{upt,G}$ and $J_{upt,j}$, respectively (Fig. 1a). Each strain produces the 'donated' metabolite $i$ at $J_{leak,i}$, which is proportional to glucose uptake with a constant proportionality of $\phi_i$. This constant represents the proportion of glucose flux going to metabolite overproduction rather than growth. Strains grow at a rate $J_{grow}$, which is a function of the uptake rate of the growth-limiting metabolite (either the 'receiver' metabolite or glucose, modified by $\phi_i$). See Methods for full details of model structure and full derivation.

We initially explored the impact of interactions in the system by simulating a nominal parameter set (Supplementary Note 1). Varying metabolic overproduction (by varying $\phi_i$ and its impact $J_{leak,j}^{y_i}$ in Fig. 1a) shows that different production rates lead to different timings and sizes of metabolite peaks, which is crucial for co-culture design (Fig. 1b left). The model reveals a nonlinear relationship between growth rate and metabolite production, with a peak of production corresponding to a metabolite production leak of 50% ($\phi_i = 0.5$) (Fig. 1b right). We extended the model to that of a co-culture system composed of two strains, denoted $i = 1$ (producing metabolite 2) and $i = 2$ (producing metabolite 1), as depicted in Fig. 1a. We simulated this system to gain an understanding of how metabolite exchange impacts co-culture dynamics. The model demonstrates that high populations are only achieved at intermediate metabolite production (that is, intermediate $\phi_1$ and $\phi_2$ values where glucose flux is evenly divided between exchange metabolite production and growth). Excess metabolite 'donations' aid receiver cell growth but at the expense to donor cells (Fig. 1c). At low $\phi_1$ and $\phi_2$ (that is, most glucose flux goes to growth), metabolite production rates are not sufficient to support growth of both strains (Fig. 1c). Asymmetric production rate (for example, $\phi_1 \gg \phi_2$ or vice versa where one strain produces excess metabolite at the expense of its own growth) can support good growth at a skewed ratio: the high-production strain supports large growth of the poor producer, which in turn generates enough metabolites to support the smaller, productive population (Fig. 1d). These dynamics result in a 'horseshoe' where large populations are obtainable at low $\phi$ when the values are similar or where there is a large difference between $\phi_1$ and $\phi_2$ (Fig. 1c).

Co-culture systems are composed of multiple nonlinear processes (including metabolite production and growth) and natural feedback effects (for example, metabolite overproduction leads to a nonlinear effect on growth rate). The dynamics of these processes are governed by the system's parameters, such as nutrient assimilation and production rates. To understand how each parameter influences the behaviour of co-culture systems, we took an ensemble modelling approach using global sensitivity analysis. In this approach, we simulated with parameters drawn evenly across biological ranges. We then assessed what impact variation in each parameter has on the variation of a given performance metric (for example, batch culture time, final population ratio). Performance is 'sensitive' to a given parameter when varying that parameter results in a large change in the metric. The global approach utilized in this work concurrently explores relationships in multiple parameter contexts. The first-order index is the direct impact varying a parameter has on the metric, while the total sensitivity is the impact of the parameter and any interactions it has with other parameters due to the model's underlying structure (see Methods for a further description). We assessed the sensitivity of the following key metrics of co-culture dynamics: final total population, batch culture time, final population composition, growth rate of each strain and metabolite uptake and production rates.

Analysis of two-member co-cultures revealed that final population size is most sensitive to the metabolite exchange parameters ($\phi_i$) but relatively insensitive to other experimentally tractable dials such as metabolite supplementation ($x_{0,i}$) and initial population ratios ($r_{0,i}$) (Fig. 1e). Batch culture times are most sensitive to experimentally intractable glucose accumulation parameters ($V_{max,G}^{y_i}$), but the next

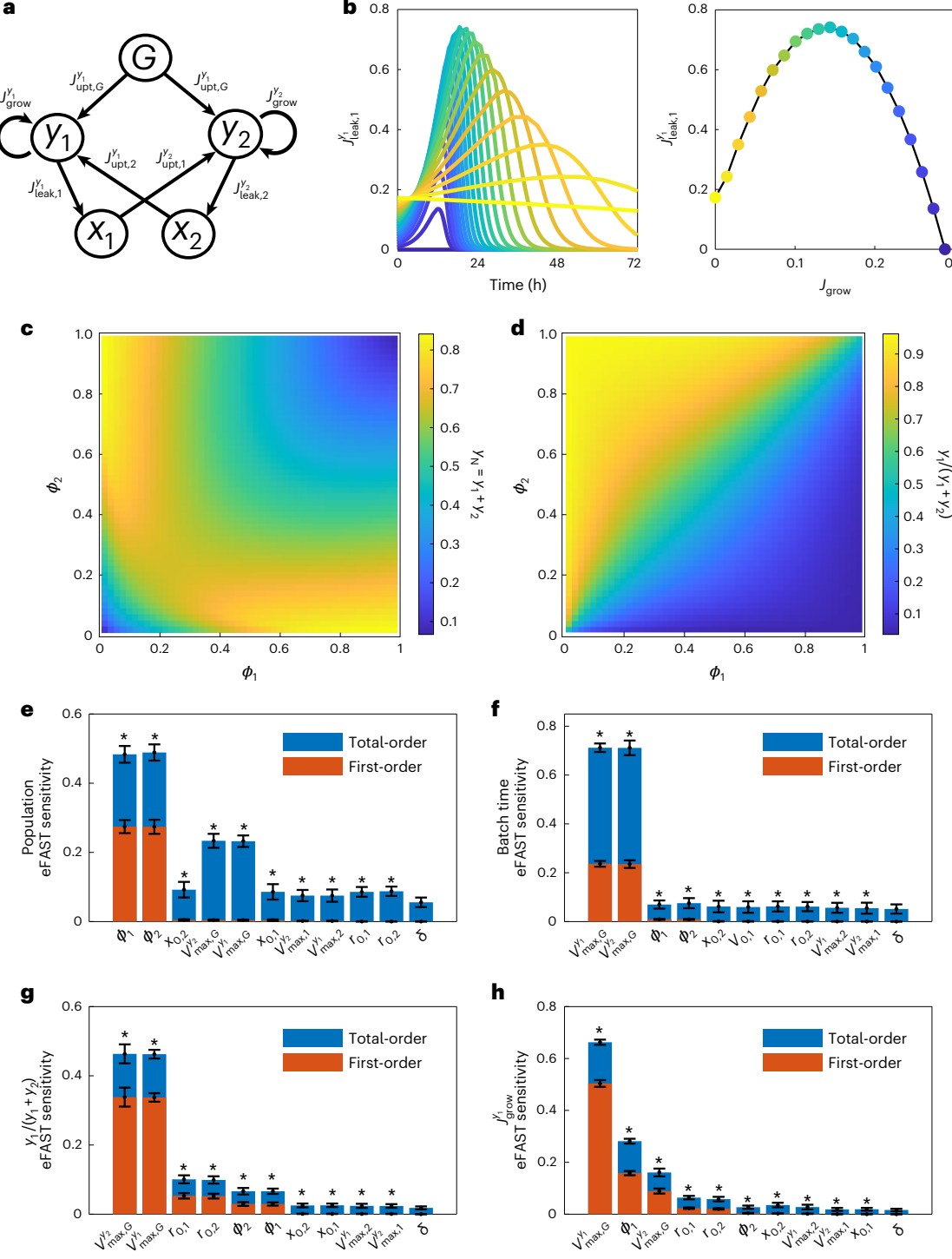

**Fig. 1 | Global sensitivity analysis of synthetic co-cultures. a**, Cartoon depiction of the model (full description in Methods). *G* is the culture carbon source (for example, glucose), $x_i$ is the essential metabolite produced/received by the co-culture system and $y_i$ is the population of strain *i*. **b**, Initial simulations of the impact of metabolite production on host growth in yeast monocultures. Left: metabolite production rate over time. Right: metabolite production has a nonlinear relationship with maximal growth. Colours represent the strength of the $\phi_1$ parameter, which governs the production of the exchanged metabolite $x_1$. **c,d**, Simulations of the two-member co-culture system at different strengths of metabolite exchange. **c**, Total population size at 72 h. **d**, Proportion of strain 1, $y_1$, in the culture. **e–h**, Global sensitivity analysis of the two-member co-culture. Model is described in Methods and results are fully discussed in Supplementary Note 2. Parameters are as follows: $\phi_i$ is the proportion of glucose flux going to

production of metabolite *i* by strain $y_i$. $x_{0,i}$ is the initial concentration of metabolite *i* in the medium. $r_{0,i}$ is the initial starting population of strain *i* (note that $r_{0,1} + r_{0,2} = 1$). $V^{y_i}_{\max,G}$ is the maximum uptake rate of glucose *G* by strain $y_i$. $V^{y_i}_{\max,j}$ is the maximum uptake rate of metabolite *j* by strain $y_i$. $\delta$ is the dummy parameter used for statistical tests in the local sensitivity analysis as described in Methods. Asterisk denotes either sensitivity or total sensitivity is significantly different ($P < 0.01$) from the dummy parameter as determined by a *t*-test using Bonferroni correction. Results are reported as mean ± s.d. for 100 resamplings. Shown are the sensitivities of the final OD$_{700}$ (**e**), the total batch culture time (**f**), the final population ratio (**g**) and the growth rate of each strain (**h**) to key parameters in the model.

most sensitive parameters are tractable metabolite exchange (Fig. 1f). Final population composition is sensitive to tractable parameters including initial population ratio and the metabolite exchange rates ($\phi_i$). The growth rate of each strain, $y_i$, is determined primarily by its own glucose assimilation rate (with 50% of the sensitivity corresponding to $V^{y_i}_{\max,G}$) (Fig. 1g,h). The remaining control of $y_i$ growth rate is shared across the starting population ratios ($r_{0,1}$ and $r_{0,2}$), glucose assimilation of the partner strain (that is, $V^{y_j}_{\max,G}$, where $j \neq i$) and the metabolite production rate $\phi_i$, showing again that overproduction of the exchange metabolite is a key driver of population dynamics (Fig. 1g).

Our full global sensitivity analysis (summarized in Supplementary Notes 2 and 3) suggests that control of co-culture dynamics is spread across few parameters within the system. A large portion of the control is spread across parameters that are difficult to experimentally engineer (for example, biomass production, glucose and metabolite assimilation), but initial population ratio and metabolite exchange rate may exert sufficient influence to control co-culture dynamics. Therefore, we focused on these two 'engineerable dials' for further experimental exploration.

### Building a toolkit for establishing synthetic co-cultures

To create microbial communities with predefined growth dynamics, we sought to modulate the production and exchange of essential metabolites between auxotrophic and overproducer strains. The most widely used yeast strain BY4741 (MATa *his3Δ1 leu2Δ0 met15Δ0 ura3Δ0*) has four auxotrophic markers including histidine (His), leucine (Leu), methionine (Met) and uracil (ura). BY4741 strains can be rendered prototrophic when harbouring episomal genetic elements (for example, pHLUM v.2 plasmids[35]) that express the *His3*, *Leu2*, *Met15* and *Ura3* genes. Under these conditions, the genome-residing auxotrophic markers in BY4741 become inconsequential, and thus these loci can instead be viewed as discrete 'modules', which can either be remediated or replaced with alternative genetically encoded 'parts.' Therefore, we chose BY4741 as a baseline from which to develop new strains, where various combinations of genes (for example, fluorescent markers or bioproduction pathways) can be integrated across four genomic loci. We created 'donor' and 'receiver' phenotypes for the generation of customized yeast communities (Fig. 2), which are compatible with the widely used modular cloning yeast toolkit (YTK)[36] and the yeast prototrophy kit[35]. Three fluorescence proteins (sfGFP, mTagBFP2 and mScarlet-I[36]) were chosen as markers to track microbial population, as they had very limited effect on cell growth and biomass across different nutritional media (Supplementary Fig. 11).

We first established cross-feeding BY4741 co-cultures with adenine-lysine (ade-Lys) and leucine-tryptophan (Leu-Trp) auxotrophic/overproducer pairs. Others have previously demonstrated their use for syntrophic communities[24,25]. These co-cultures showed significantly higher cell growth than their corresponding monocultures (Supplementary Fig. 12). We next created additional cross-feeding BY4741 co-cultures by first reviewing amino acid and nucleotide biosynthesis pathways[37] and then selected genes that would overproduce amino acids and nucleotides when overexpressed. We chose *ade4op*[25,38], *ura4*, *his1*[39,40], *trp2Fbr*[41], *aro3Fbr*[42,43], *aro4Fbr*[44,45], *aro7Fbr*[45,46], *leu4Fbr*[24,47], *ilv6 G89D*[48], *mpr1 G85E*[49], *lys21op*[50], *ser2*, *cys3*, *met6* and *hom3-R2*[51] (Supplementary Table 1). We also created the reciprocal auxotrophic strains or obtained them from the Yeast Knockout Library[52]: *ade8Δ*, *ura3Δ*, *his3Δ*, *trp1Δ*, *tyr1Δ*, *pha2Δ*, *aro7Δ*, *leu2Δ*, *ilv1Δ*, *arg4Δ*, *lys2Δ*, *ser1Δ*, *cys4Δ*, *met14Δ* and *thr4Δ* (Supplementary Tables 1 and 2, and Fig. 13).

We first assessed whether the newly generated auxotrophic and overproducing strains could establish cross-feeding co-cultures with adenine auxotrophs (*ade8Δ*). Adenine auxotrophs exchanged adenine at either nominal or increased levels (*ADE4op* overexpression) and were paired with other auxotrophs also expressing an exchangeable metabolite at nominal or increased levels, for a total of 52 co-cultures

(Fig. 2). See Supplementary Note 5 and Extended Data Fig. 1 for full details. On the basis of the growth (optical density at 700 nm (OD$_{700}$)) of the co-cultures, we classified each target metabolite by their ability to facilitate growth in cross-feeding co-cultures: strong (OD$_{700} \geq 0.5$): adenine, Trp, Met, His; medium ($0.3 \leq$ OD$_{700} < 0.5$): Lys, Phe+Tyr, Val+Ile, Cys, Leu, Ura; and weak (OD$_{700} < 0.3$): Thr, Tyr, Arg, Ser. In some cases (ade-His, ade-Lys, ade-Phe+Tyr, ade-Thr, ade-Trp), the overexpression of the target metabolite improved co-culture growth as predicted (Extended Data Fig. 1). We then performed LC–MS to confirm that the overexpression of target genes (chosen for the overproduction of adenine, His, Lys, Phe, Tyr, Trp, Thr) indeed enhanced the production of their corresponding metabolite (Supplementary Fig. 17). Thus, the molecular toolkit includes 3 fluorescence proteins, 15 auxotrophies (13 presenting strict auxotrophic phenotypes) and 15 exchanged metabolites (7 whose level can be modulated by gene overexpression in the tested conditions). The toolkit can be used for the development of novel cross-feeding co-cultures by exploring their combinations.

### Designing synthetic two- and three-member co-cultures

We used auxotrophic/overproducer strains from our toolkit to create additional syntrophic co-cultures composed of two or three members. Having previously validated co-cultures with adenine, we arbitrarily decided to test co-cultures with Lys (which performed well in the ade-Lys co-culture). We established His-Lys, Leu-Lys, Phe-Lys, Trp-Lys^v1, Trp-Lys^v2, Val-Lys^v1 and Val-Lys^v2 co-cultures (described in Fig. 3 and its caption), which displayed significantly higher cell growth than their monoculture controls (Fig. 3a–c). We then extended upon a subset of these two-member co-cultures (adding additional adenine, Lys, Trp and His targets) to create 5 pairs of three-member co-cultures, which exhibited one-way communication (where each member presents one auxotrophy). These three-member co-cultures were named AKW_I, AKW_II, AKH_III, AKM_IV and HKM_V (acronyms to denote their component auxotrophs and overexpressed metabolite targets; I, II, III, IV, V refer to co-culture number), and their controls included the monocultures of each member and all combinations of two-member co-cultures (Fig. 3d,e). All monoculture controls showed limited growth (as expected for essential metabolite auxotrophs; Supplementary Fig. 19). Some two-member co-culture controls did exhibit different degrees of growth, including AK_I, AW_I, KW_I, KW_II, AK_II, AW_II and AK_IV. Growth observed from these two-member systems is probably due to unanticipated cross-feeding behaviour from leaky secretion of an additional cross-feeding metabolite, in addition to the expected one. For example, in the two-member control AK_I, which consisted of member A_I (*trpΔ*ade+) and K_I (*adeΔ*Lys+), it is expected that A_I would complement K_I by secreting adenine but K_I was not expected to complement A_I ('+', overexpression of target metabolite). However, the clear growth of this co-culture suggests that K_I may 'leak' enough Trp to complement A_I (that is, the strain naturally secretes a low level of Trp) (Fig. 3d–f). We were able to quantify Trp levels in the supernatant even when no specific tryptophan synthesis gene was overexpressed, explaining this result (Supplementary Fig. 17). All three-member co-cultures via one-way communication showed significant cell growth compared with corresponding monoculture and two-member co-culture controls. Three of the three-member co-cultures showed strong cell growth (AKW_I, AKW_II, AKM_IV), and two showed weak cell growth (AKH_III, HKM_V) (Fig. 3f and Extended Data Fig. 2).

We next developed three-member co-cultures that operated with two-way communication (where each strain has two auxotrophies), using the targets ade, Lys, Trp, His and Met, which we labelled AKW_VI, AKH_VII, AKM_VIII and HKM_IX (Fig. 3d–f). The co-cultures AKW_VI and AKM_VIII had significantly higher cell growth compared with their controls of monocultures and two-member co-cultures; however, AKH_VII and HKM_IX did not follow this trend. As expected, the controls of monocultures did not grow because these strains were auxotrophic to two essential metabolites. Unlike the controls of

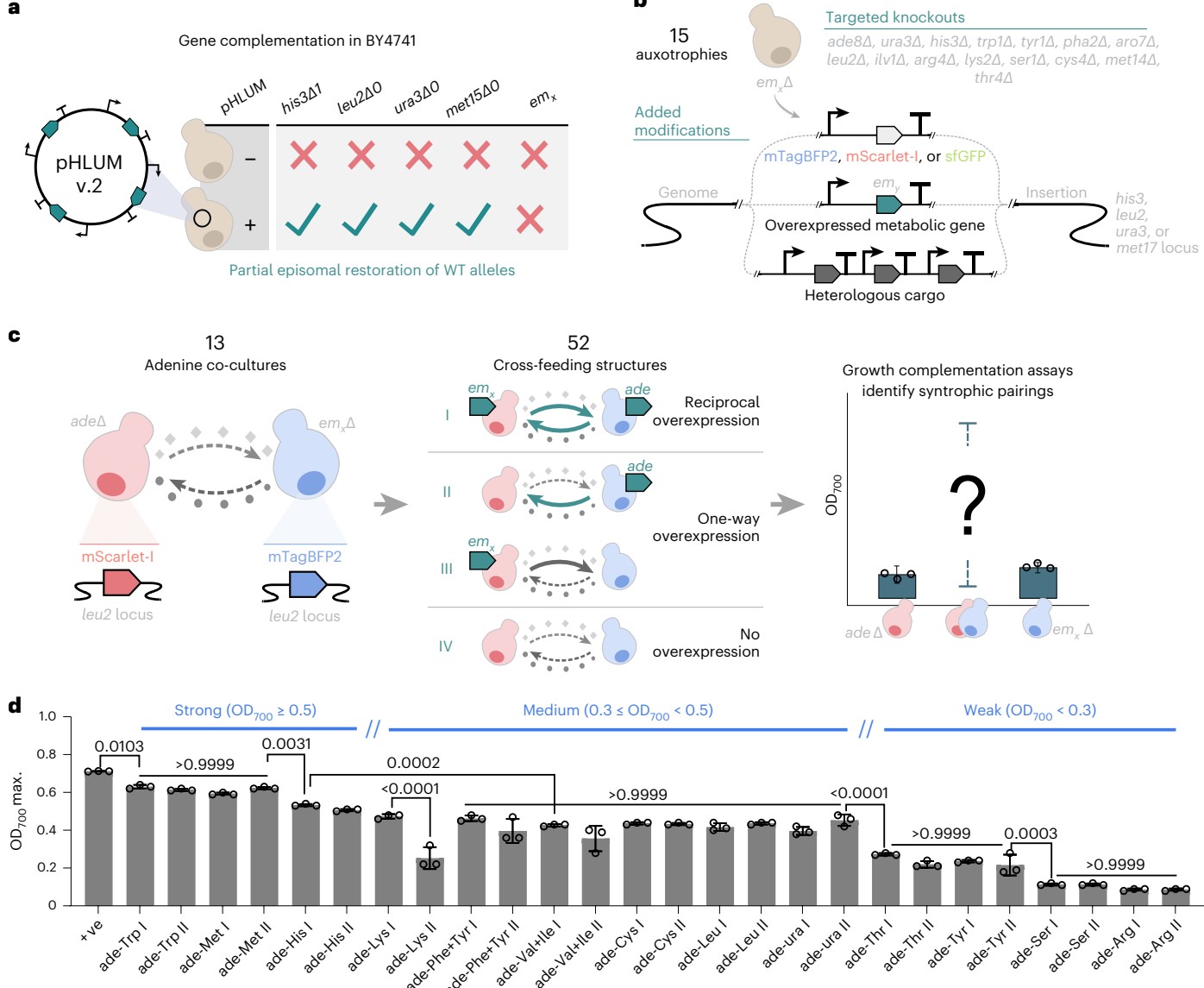

**Fig. 2 | Modularity of yeast *S. cerevisiae* for co-culture toolkit development.**
**a**, Model yeast *S. cerevisiae* BY4741 has four auxotrophic markers of histidine, leucine, uracil and methionine, and the yeast prototrophy kit (pHLUM v.2 plasmids)[35] can be used to complement the auxotrophic markers. **b**, Auxotrophic markers in BY4741 can be replaced with functional modules that facilitate co-culture design. We developed 15 knockout strains: *ade8Δ*, *ura3Δ*, *his3Δ*, *trp1Δ*, *tyr1Δ*, *pha2Δ*, *aro7Δ*, *leu2Δ*, *ilv1Δ*, *arg4Δ*, *lys2Δ*, *ser1Δ*, *cys4Δ*, *met14Δ* and *thr4Δ*, which are auxotrophic to adenine, uracil, histidine, tryptophan, tyrosine, phenylalanine, tyrosine and phenylalanine, leucine, valine and isoleucine, arginine, lysine, serine, cysteine, methionine and threonine, respectively (Supplementary Table 1). One marker such as *leu2Δ* can be used to express different fluorescent proteins such as mTagBFP2, mScarlet-I and sfGFP, which can be used as fluorescent markers. One marker such as *his3Δ* can be used to express the genes that help the production of exchanged metabolites (em_y), which can be

used as metabolite donor. One marker such as *met15Δ* or CRIPSR-cas9 tool can be used to express a heterologous high-value bioproduct synthesis pathway. **c**, An adenine auxotrophic strain was designed to pair with 13 other auxotrophic strains for co-culture potential in 4 different cross-feeding structures (ade-em I, ade-em II, ade-em III, ade-em IV). The auxotrophs for each metabolite, with and without adenine overproduction (*ade4op* overexpression) were co-cultured with the adenine auxotroph (*ade8Δ*) with and without the overexpression of genes involved in the overproduction of the metabolites to identify the syntrophic pairs. **d**, The maximal OD_700 values of co-culture ade-em I and ade-em II within 72 h were ranked from strong to weak. N = 3 biologically independent samples and data are presented as mean ± s.d. One-way ANOVA, followed by Bonferroni's multiple comparisons test with 95% confidence intervals were performed using GraphPad Prism 9.5.0 software and P values are noted.

two-member co-cultures via one-way communication, most controls of two-member co-cultures via two-way communication also did not show obvious cell growth due to leakage, which suggests that this is a good strategy when a tighter control over cross-feeding is desired. The marked increase in cell growth observed in AKW_VI and AKM_VIII suggests a high level of interdependence among all three members in the three-member co-culture system.

## Population-controlling strategies in synthetic co-cultures
The global sensitivity analysis identified metabolite exchange as a key determinant of population size and batch culture time (Fig. 1). Therefore, we next engineered the promoters of genes encoding the metabolic enzymes responsible for the overproduction of cross-feeding metabolites to tune overall metabolite exchange strength. Five promoters with different strengths (pCCW12, pTEF1, pRL18B, pPOP6, pREV1)

**a**

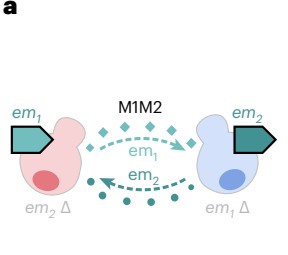

**b**

| Two-member Co-culture | Strain_em1_Red | Strain_em2_Blue |
|---|---|---|
| Description | Phenotype | Phenotype |
| His-Lys | HisΔLys+ | His+LysΔ |
| Leu-Lys | LeuΔLys+ | Leu+LysΔ |
| Phe-Lys | PheΔLys+ | Phe+LysΔ |
| Trp-Lys$^{v1}$ | TrpΔLys+$^{v1}$ | Trp+LysΔ$^{v1}$ |
| Trp-Lys$^{v2}$ | TrpΔLys+$^{v2}$ | Trp+LysΔ$^{v2}$ |
| Val-Lys$^{v1}$ | ValΔLys+$^{v1}$ | Val+LysΔ$^{v1}$ |
| Val-Lys$^{v2}$ | ValΔLys+$^{v2}$ | Val+LysΔ$^{v2}$ |

**c**

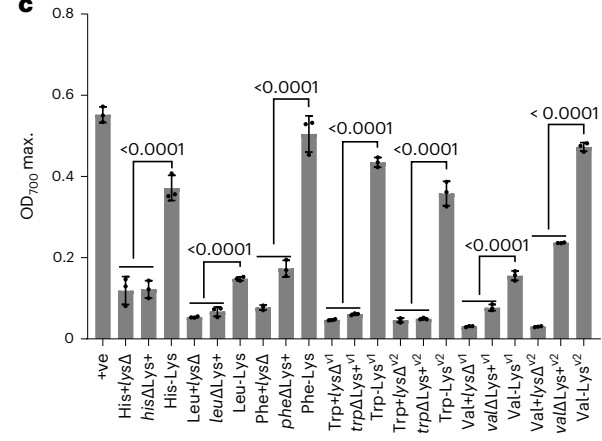

**d**

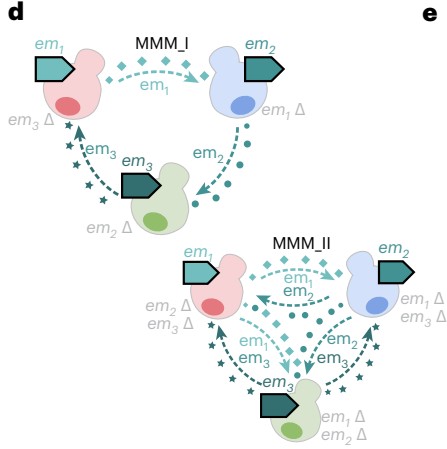

**e**

| Three-member co-culture | | Strain_em1_Red | | Strain_em2_Green | | Strain_em3_Blue | |
|---|---|---|---|---|---|---|---|
| Description | No. | Phenotype | No. | Phenotype | No. | Phenotype | No. |
| ade-Lys-Trp | AKW_I | *ade*ΔLys+ | K_I | *trp*Δade+ | A_I | *lys*ΔTrp+ | W_I |
| ade-Lys-Trp | AKW_II | *ade*ΔTrp+ | W_II | *trp*ΔLys+ | K_II | *lys*Δade+ | A_II |
| ade-Lys-His | AKH_III | *ade*ΔLys+ | K_III | ade+*his*Δ | A_III | *lys*ΔHis+ | H_III |
| ade-Lys-Met | AKM_IV | *ade*ΔLys+ | K_IV | ade+*met*Δ | A_IV | *lys*ΔMet+ | M_IV |
| Lys-His-Met | HKM_V | *met*ΔLys+ | K_V | Met+*his*Δ | M_V | *lys*ΔHis+ | H_V |
| ade-Lys-Trp | AKW_VI | *ade*ΔLys+*trp*Δ | K_VI | ade+*lys*Δ*trp*Δ | A_VI | *ade*Δ*lys*ΔTrp+ | W_VI |
| ade-Lys-His | AKH_VII | *ade*ΔLys+*his*Δ | K_VII | ade+*lys*Δ*his*Δ | A_VII | *ade*Δ*lys*ΔHis+ | H_VII |
| ade-Lys-Met | AKM_VIII | *ade*ΔLys+*met*Δ | K_VIII | ade+*lys*Δ*met*Δ | A_VIII | *ade*Δ*lys*ΔMet+ | M_VIII |
| Lys-His-Met | HKM_IX | Lys+*his*Δ*met*Δ | K_IX | *lys*Δ*his*ΔMet+ | M_IX | *lys*ΔHis+*met*Δ | H_IX |

**f**

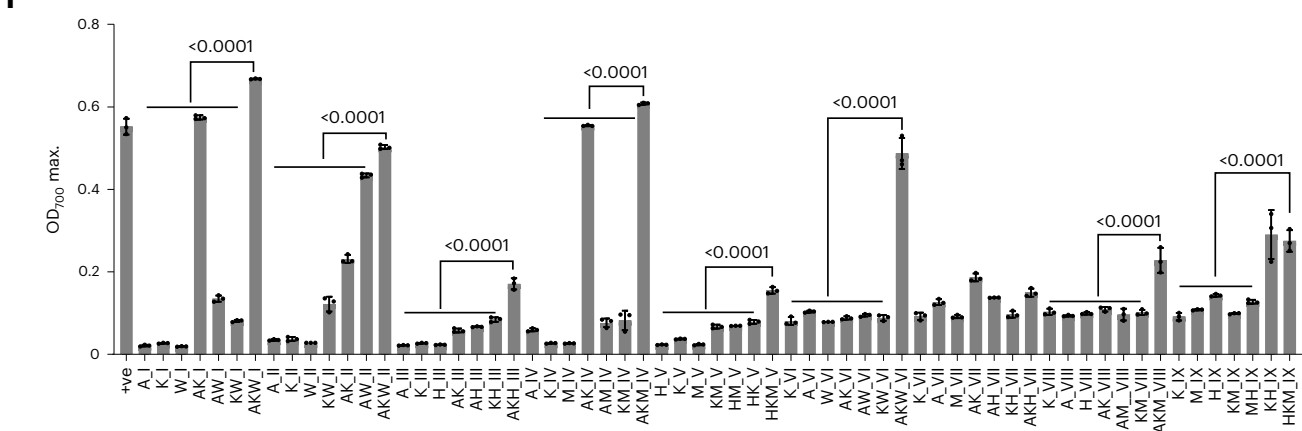

**Fig. 3 | Rational design of synthetic two- and three-member co-cultures.**
**a**, Diagram of two-member cross-feeding co-cultures M1M2. The two members were labelled with blue fluorescent protein mTagBFP2 and red fluorescent protein mScarlet-I, respectively. Each member is auxotrophic to one exchanged metabolite (em1) and overproduced another exchanged metabolite (em2). **b**, The phenotype of each member and the strain combination for co-cultures are listed in the table. Val-Lys$^{v1}$, cell culture using SM medium; Val-Lys$^{v2}$, cell culture using SM plus isoleucine; Trp+LysΔ$^{v1}$ (BFP-tagged) contains the native Trp pathway; Trp+LysΔ$^{v2}$ (BFP-tagged) does not contain the native Trp pathway. **c**, Maximal OD$_{700}$ values of two-member co-cultures and monoculture controls within 72 h. $N = 3$ biologically independent samples and data are presented as mean ± s.d. One-way ANOVA, followed by Bonferroni's multiple comparisons test with 95% confidence intervals were performed using GraphPad Prism 9.5.0 and $P$ values are noted. **d**, The diagram of three-member co-cultures via one-way (top) and two-way (bottom) communication; each member was labelled with one fluorescent protein mScarlet-I, mTagBFP2 or sfGFP, respectively. In one-way-communicated three-member co-cultures (MMM_1), each member is auxotrophic to one

exchanged metabolite (em1 or em2) and overproduced another exchanged metabolite (em2 or em3). In two-way-communicated three-member co-cultures (MMM_2), each member is auxotrophic to two exchanged metabolites (em12, em13 or em23) and overproduced another exchanged metabolite (em3, em2 or em1). **e**, The strain combination table of 9 pairs of three-member co-cultures. We labelled three-member co-cultures and controls (monoculture and two-member co-cultures) using target gene abbreviations. For example, in three-member co-culture ade-Lys-Trp (labelled as AKW_I), monoculture controls of each member of *ade*ΔLys+ (RFP-tagged), *trp*Δade+ (BFP-tagged), *lys*ΔTrp+ (GFP-tagged) are labelled as K_I, A_I and W_I; controls of two-member co-cultures are labelled as AK_I, AW_I, KW_I. **f**, Maximal OD$_{700}$ values of the three-member co-cultures and the controls of monocultures and two-member co-cultures within 72 h. In these two- and three-member co-cultures, the initial OD$_{700}$ was 0.078 for each member, and the initial ratios were 1:1 and 1:1:1, respectively. $N = 3$ biologically independent samples and data are presented as mean ± s.d. One-way ANOVA, followed by Bonferroni's multiple comparisons test with 95% confidence intervals were performed using GraphPad Prism 9.5.0 and $P$ values are noted.

were used to modulate the expression of target enzymes among 5 pairs of two-member cross-feeding co-cultures: ade-Lys (Fig. 4), along with Leu-Trp, Val-Lys, Trp-Lys and His-Lys (Supplementary Figs. 20–23). As predicted by the model, varying promoter strength had a significant impact on both batch culture time and growth. Co-culture growth and the population fraction tagged with RFP were positively correlated with the promoter strength of *ade4op* even under various promoter strengths of *lys21op* (Fig. 4e,f). *Lys21op* expression appeared to benefit the co-culture growth only under strong promoters such as *pCCW12* and *pTEF1* (Fig. 4e), coinciding with a reduced fraction of RFP-tagged population (the '*ade*ΔLys+' strain), especially when *ade4op* was weakly expressed. The *ade*ΔLys+ strain (tagged with RFP) became dominant in ade-Lys co-culture combinations, and populations with stronger promoters had both shorter log phases and higher cell growth (Fig. 4g). Overall, we observed that altering the promoter strength of enzymes that contributed to metabolite exchange could steer co-culture growth and population fractions across different pairs (Supplementary Figs. 20–23).

We next tested how initial population ratios influenced cell growth and population size over time for four pairs of co-cultures (ade-Tyr, ade-Phe, ade-Val, ade-Arg), where each co-culture pair displayed different growth dynamics when the initial ratio was 1:1 (Extended Data Figs. 1 and 3). Three initial ratios were selected to test on these co-cultures: 10:1, 1:1 and 1:10. In the ade-Tyr co-culture, the initial ratio 10:1 showed higher co-culture growth compared with initial ratios 1:1 and 1:10. The *ade*ΔTyr+ population (tagged with RFP) was the dominant community member and surpassed the ade+*tyr*Δ population (tagged with BFP) even when the co-culture started with a lower fraction of *ade*ΔTyr+ such as 10:1. Interestingly, the ade-Tyr co-culture performed much better at 10:1 ratio (Extended Data Fig. 3c–e). Each community member in the ade-Phe and ade-Val co-cultures showed robust growth, with population ratios being controlled by the initial ratio (with blue dominating at 10:1, red at 1:10 and equal proportions at 1:1 (Extended Data Fig. 3f–k)). The co-culture ade-Arg did not have observable growth under these three initial ratios (Extended Data Fig. 3l–n), which was consistent with the 1:1 ratio (Extended Data Fig. 1).

We then evaluated the effects of varying exchanged metabolite (em) supplementations on the growth and population size of synthetic co-cultures, which included 3 two-member co-cultures (ade-Lys, His-Lys, Trp-Leu; Supplementary Fig. 24 and Extended Data Fig. 4) and 2 three-member co-cultures (AKH_III and AKW_VI) which operated on either one-way or two-way communication (Fig. 5). These strains were selected on the basis of their previously observed ability to grow well when co-cultured, which served as a suitable baseline for further modifications. The addition of either adenine, lysine or histidine to AKH_III increased the co-culture growth (Fig. 5b), which suggests that co-culture growth is still limited by cross-feeding rates. As expected, there is an increase in the ratio of *ade*ΔLys+ (RFP-tagged) in response to adenine addition, an increase in *lys*ΔHis+ (BFP-tagged) with the addition of lysine and an increase in ade+*his*Δ (GFP-tagged) with the addition of histidine. While *ade*ΔLys+ (RFP-tagged) and *lys*ΔHis+

(BFP-tagged) increased their ratio with the dosage of exchanged metabolite supplement, the opposite behaviour was found for ade+*his*Δ (GFP-tagged) (Fig. 5c). A different behaviour was observed in co-culture AKW_VI, where total growth was not affected by the supplementation with adenine, lysine and tryptophan. The two-way communication co-culture, with the double auxotrophs and the competition for the supplemented metabolite, complicates the dynamics of the system. The supplementation with lysine and tryptophan led to more significant changes of the co-culture composition than adenine supplementation (Fig. 5e,f). Compared with the AKW_VI co-culture without metabolite supplementation, adding adenine (10 mg l⁻¹) yielded up to +11.0% GFP-tagged population, −6.9% (BFP) and −4.1% (RFP); adding lysine (50 mg l⁻¹) yielded up to +7.1% (GFP), +14.1% (BFP) and −21.2% (RFP); and adding tryptophan (10 mg l⁻¹) yielded up to −6.8% (GFP), −30.3% (BFP) and +37.1% (RFP) (Fig. 5f).

We next tested different initial cell densities (OD$_{700}$ 0.067, 0.078, 0.102, 0.148) for 4 pairs of three-member co-cultures via two-way communication, including AKW_VI, AKH_VII, AKM_VIII and HKM_IX. We found that with higher initial OD$_{700}$ values, higher co-culture growth can be achieved (Fig. 5g and Supplementary Figs. 27–31). Moreover, the growth dynamics of each member were distinct under different initial cell densities. Taking co-culture AKW_VI as an example, when the initial OD$_{700}$ was 0.067, populations tagged with BFP (strain *ade*Δ*lys*ΔTrp+) and RFP (strain *ade*ΔLys+*trp*Δ) displayed comparable cell growth, which was higher than the GFP-tagged population (strain ade+*lys*Δ*trp*Δ) within 72 h. When the initial OD$_{700}$ was 0.078, the population tagged with BFP (strain *ade*Δ*lys*ΔTrp+) became higher than the population tagged with GFP (strain ade+*lys*Δ*trp*Δ) at ~40 h, followed by the population tagged with RFP (strain *ade*ΔLys+*trp*Δ). The population tagged with RFP (strain *ade*ΔLys+*trp*Δ) became dominant in some growth periods when the initial OD$_{700}$ was 0.102 and 0.148 (Fig. 5g). These results indicate that initial cell density could be used as a strategy to control cell growth and, to a certain extent, population size.

## Synthetic co-cultures for improved resveratrol production

Dividing metabolic pathways between multiple strains in a co-culture can sometimes increase product formation due to division of labour between the members of the communities. Therefore, we tested our toolkit for a metabolic engineering application utilizing the high-value antioxidant resveratrol as a case study[53,54]. The resveratrol synthesis pathway consists of three genes *FjTAL*, *At4CL1* and *VvVST*[55], which can easily be split into two modules[54]: one containing *FjTAL* (catalysing L-tyrosine to p-coumaric acid) and the other containing *At4CL1* and *VvVST* (catalysing p-coumaric acid to resveratrol) (Fig. 6a). Three pairs of promising cross-feeding two-member co-cultures were selected (adeLys, Trp-ade, Trp-Lys), and each member in these co-cultures was engineered with either *FjTAL* or both *At4CL1* and *VvVST*. We constructed six pairs of cross-feeding two-member co-cultures with division of labour for resveratrol production: AK_Res1, 2 (2x ade-Lys), AW_Res1, 2 (2x Trp-ade) and WK_Res1, 2 (2x Trp-Lys). Each cross-feeding pair carried out the pathway in the two possible orientations: (*FjTAL*)–(*At4CL1*

**Fig. 4 | Promoter engineering controls the growth and population size in two-member co-cultures. a**, Diagram of two-member cross-feeding co-cultures (em1-em2); BFP-tagged member is overexpressing em1 and auxotrophic to em2, RFP-tagged member is overexpressing em2 and auxotrophic to em1. Five pairs of co-cultures include ade-Lys, Leu-Trp, Val-Lys, Trp-Lys and His-Lys. Ade-Lys co-culture is used as an example here and the other 4 pairs are shown in Supplementary Figs. 20–23. **b**, The combinations of five promoters with different strengths from strong (1) to weak (5), plus 0 expression (6). The expressions of target genes *em1* and *em2* were driven by these five promoters, with no expression in the BFP- and RFP-tagged member, respectively. Then, these 6 BFP-tagged members and 6-RFP-tagged members were combined to form 36 pairs of different two-member co-cultures. **c**, The strain table for the combinations of ade-Lys two-member co-cultures. The strain is named after

the abbreviation of the target gene and the promoter strength; for example, BFP-tagged strain ade+*lys*Δ overexpressing *ADE4op* under stronger promoter *pCCW12* is abbreviated ade#1. **d**, OD$_{700}$ at 48 h of monocultures in synthetic minimal medium as the negative controls, and of the positive control (+ve, BY4741-pHLUM). *N* = 2 biologically independent samples and data are presented as mean ± s.d. **e,f**, Heat map of OD$_{700}$ values (**e**) and RFP-tagged population percentages at 48 h (**f**) of 36 pairs of ade-Lys two-member co-cultures. The initial ratio was 1:1 and the initial cell density was OD$_{700}$ 0.078 for each member in these co-cultures. *X* axis from left to right deonotes the promoter strength of Lys+ (*LYS21op*) from weak (6) to strong (1); *y* axis from bottom to top denotes the promoter strength of ade+ (*ADE4op*) from weak (6) to strong (1). **g**, Time courses of the growth of all co-cultures and each member. *N* = 2 biologically independent samples and data are presented as mean ± s.d.

and *VvVST*) denoted as 1 or (*At4CL1* and *VvVST*)–(*FjTAL*) denoted as 2 (Supplementary Fig. 32a). As an example, AK_Res2, is a co-culture of the ade-Lys pair where the ade-Lys+ strain expresses *FjTAL* and the Lys-ade+ strain expresses *At4CL1* and *VvVST*. The 12 auxotrophic monocultures

did not grow in minimal medium as expected (Supplementary Fig. 32c). We constructed a control pair of strains based on wild-type BY4741 with no cross-feeding (C_Res1; WT) and a monoculture control expressing the full resveratrol synthesis pathway (Supplementary Fig. 32a).

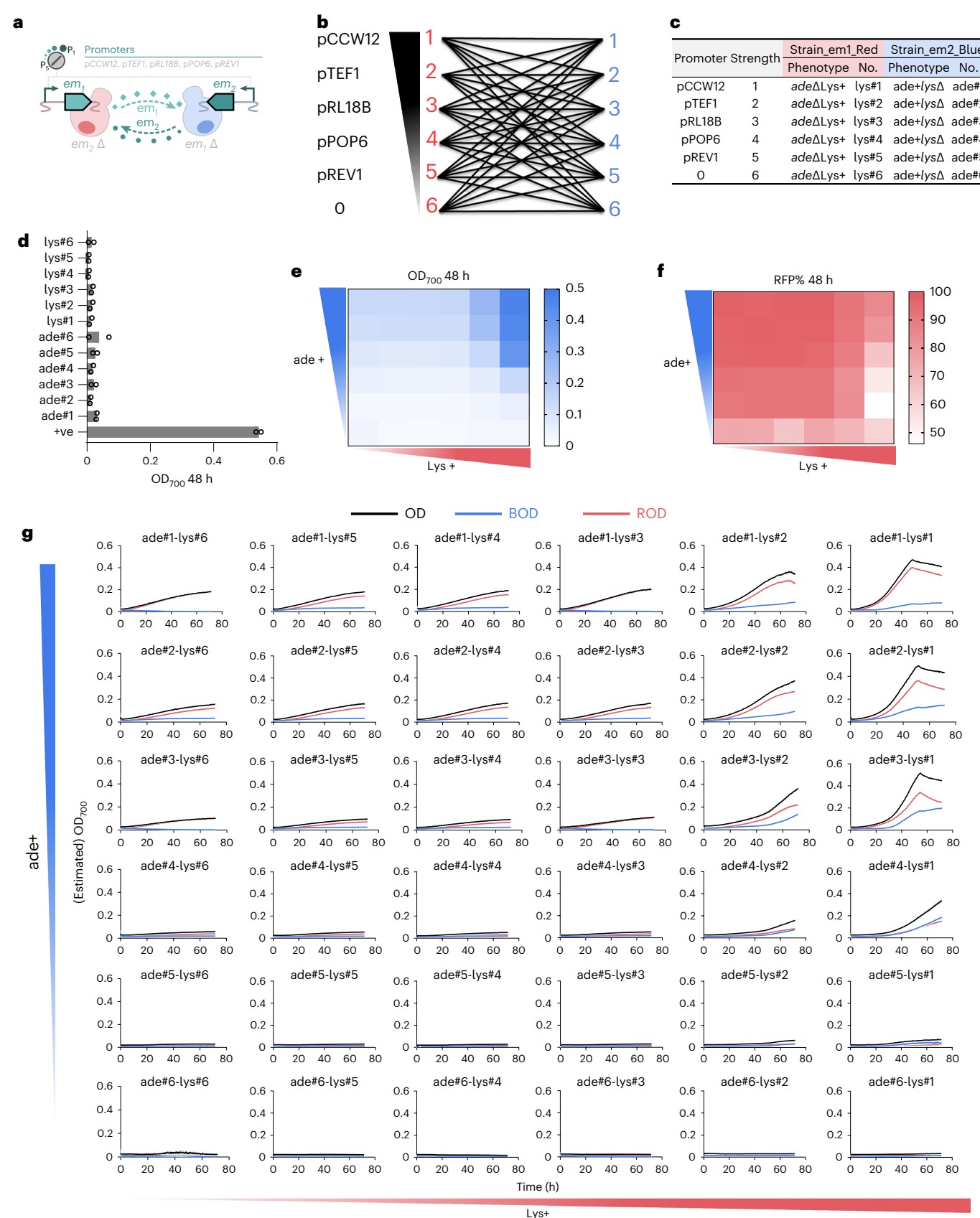

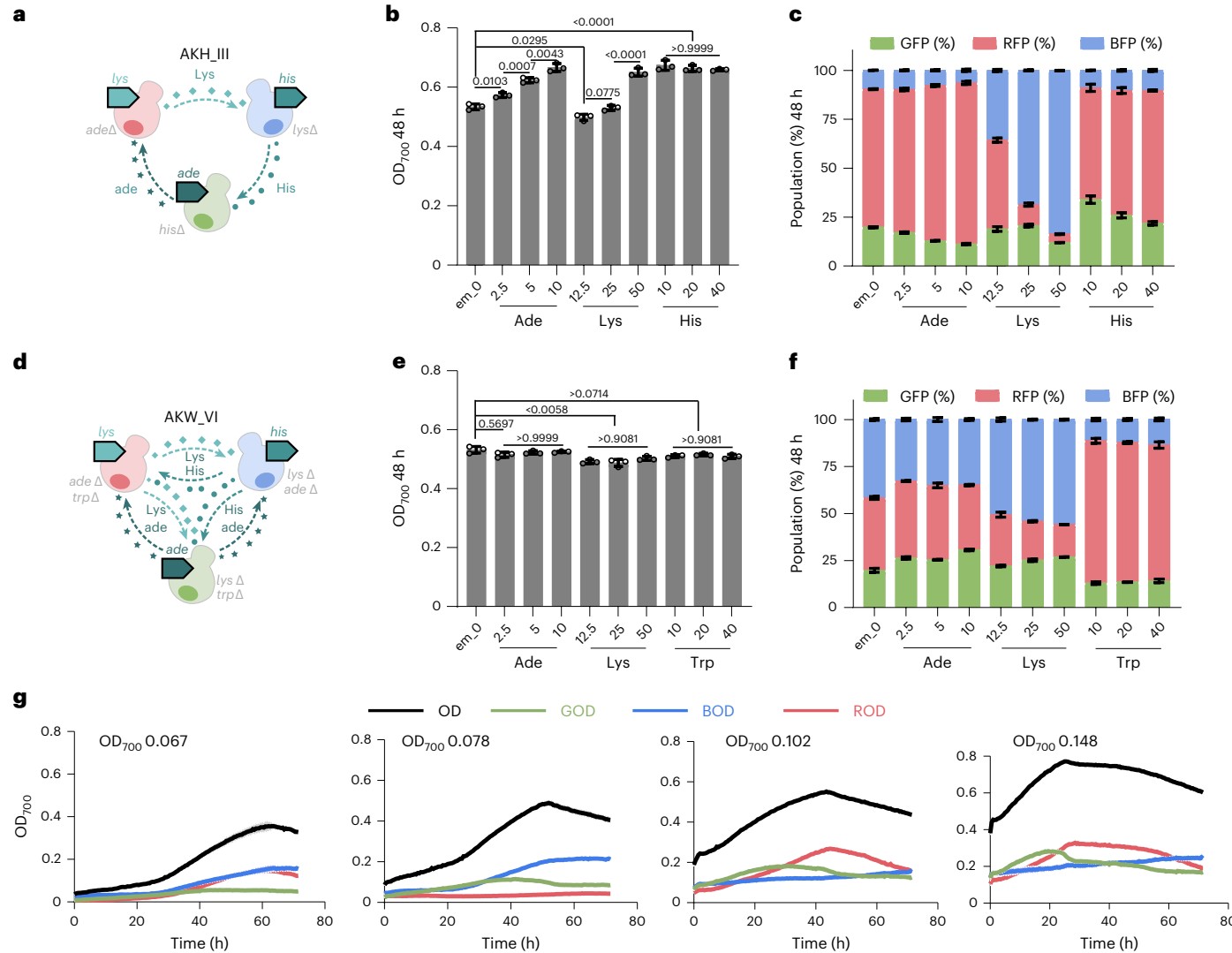

**Fig. 5 | Effect of varied metabolite supplementations and initial cell densities on growth and population size in three-member co-cultures. a**, Diagram of three-member co-culture AKH_III via one-way communication. **b,c**, The growth of co-culture (**b**) and population percentage (**c**) of each member of co-culture AKH_III with and without exchanged metabolite (em) supplementation (mg l$^{-1}$) at 48 h. em_0 means no em supplementation; the supplementation (final concentration, mg l$^{-1}$) of ade, Lys, His and Trp were 2.5, 5, 10; 12.5, 25, 50; 10, 20, 40 and 10, 20, 40, respectively. $N$ = 3 biologically independent samples and data are presented as mean ± s.d. One-way ANOVA, followed by Bonferroni's multiple comparisons test with 95% confidence intervals were performed using GraphPad Prism 9.5.0 and $P$ values are noted. **d**, Diagram of three-member co-culture AKW_VI via two-way communication. **e,f**, The growth of co-culture (**e**) and population percentage

(**f**) of each member in co-culture AKW_VI with and without em supplementation at 48 h. $N$ = 3 biologically independent samples and data are presented as mean ± s.d. One-way ANOVA, followed by Bonferroni's multiple comparisons test with 95% confidence intervals were performed using GraphPad Prism 9.5.0 and $P$ values are noted. **g**, The growth curves of co-culture AKW_VI (OD) and estimated growths of three members including GOD, BOD and ROD at different initial cell densities of OD$_{700}$ 0.067, 0.078, 0.102 and 0.148 in 72 h. OD indicates the total OD$_{700}$ values of the co-culture. GOD, BOD and ROD represent the estimated OD values for GFP-, BFP- and RFP-tagged populations, respectively. The initial ratio was 1:1:1 for each member in these co-cultures. $N$ = 3 biologically independent samples and data are presented as mean ± s.d.

We developed a dynamic model of the resveratrol co-culture and repeated our global sensitivity analysis, including an analysis of productivity and yield as well as previous performance metrics (Fig. 6b–d and Supplementary Note 4). Our analysis shows that while growth is most sensitive to $\phi_2$ (that is, production of the exchange metabolite from the resveratrol production strain), productivity and yield are most sensitive to $\phi_1$ (that is, the production of the exchange metabolite from the p-coumaric producer). These key performance metrics are equally sensitive to the starting ratio of the two strains. Therefore, we set to examine the control of resveratrol production by experimentally manipulating starting ratios. With five initial ratios of 20:1, 6:1, 1:1, 1:6 and 1:20, seven pairs of synthetic co-cultures (six with and one without

cross-feeding) were compared for cell growth, p-coumaric acid and resveratrol production at 48 h in synthetic minimal medium (Fig. 6e,f and Supplementary Fig. 32d).

We found that division of labour enables improved resveratrol production in many pairs. Co-culture AK_Res1 (at ratios 1:6, 1:20) and WK_Res1&2 show poorer growth than the monoculture control (Mctrl). However, most co-cultures showed higher OD$_{700}$ values than Mctrl, which suggests that division of labour by pathway split reduces metabolic burden. This can be seen in the control co-culture C_Res1 with division of labour but no cross-feeding whose OD$_{700nm}$ ranged from 0.46–0.53, which was significantly higher than that of Mctrl (0.44). Resveratrol titres, however, were only higher (2.6-fold) than Mctrl

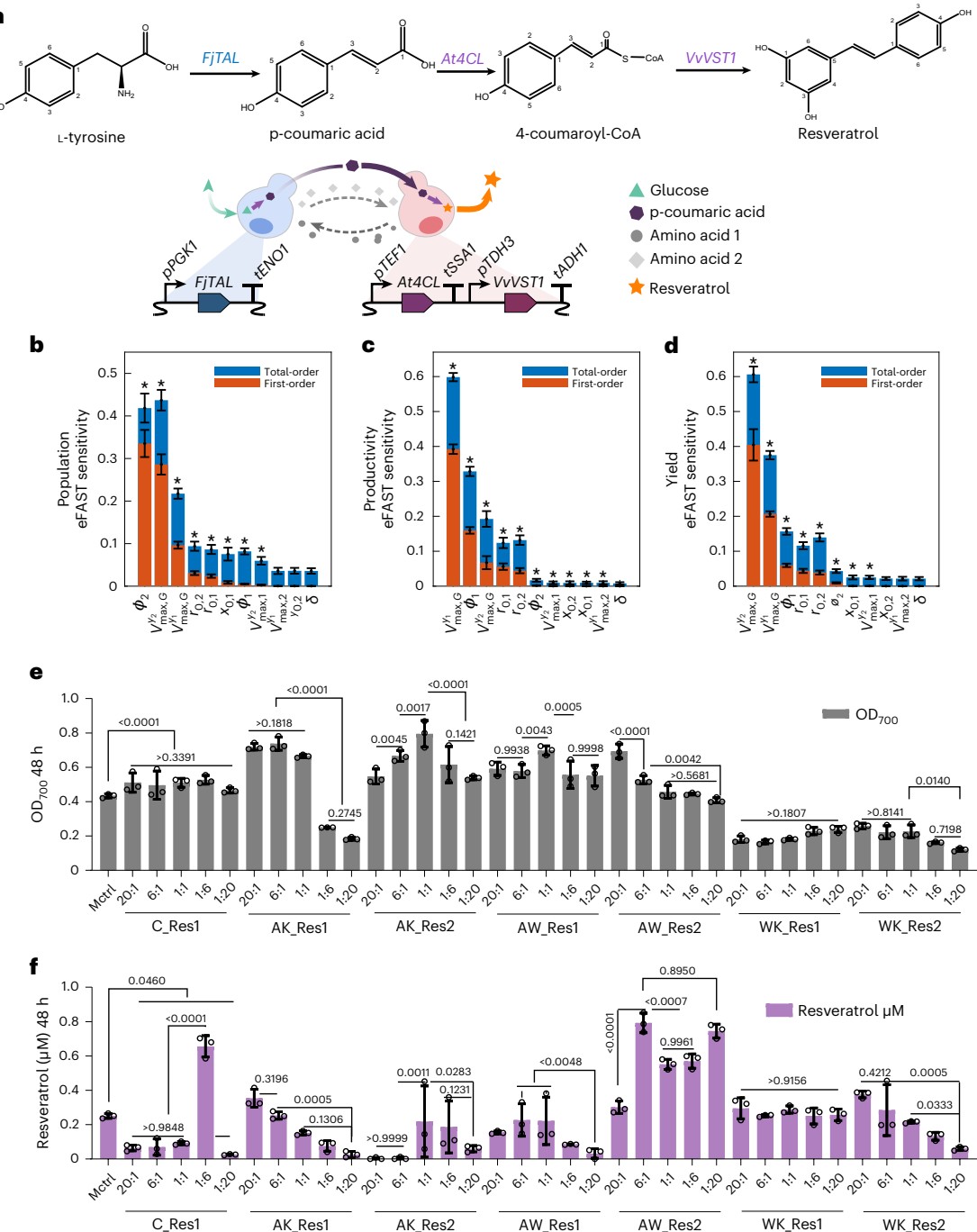

**Fig. 6 | Application of synthetic co-cultures for improved resveratrol production. a**, De novo resveratrol synthesis pathway in yeast and diagram of division of labour of resveratrol pathway in yeast cross-feeding co-cultures. **b**–**d**, Global sensitivity analysis of a division of labour biotechnological process. The model and full results are discussed in Supplementary Note 4. Model parameters are as follows: $\phi_i$, the proportion of glucose flux going to production of metabolite $i$ by strain $y_i$; $x_{0,i}$, the initial concentration of metabolite $i$ in the medium; $r_{0,i}$, the initial starting population of strain $i$ (note that $r_{0,1} + r_{0,2} = 1$); $V^{y_i}_{\max,G}$, the maximum uptake rate of glucose $G$ by strain $y_i$; $V^{y_i}_{\max,j}$, the maximum uptake rate of metabolite $j$ by strain $y_i$; $\delta$ is the dummy parameter used for statistical tests in the global sensitivity analysis as described in Methods. Shown

are the sensitivities of the final OD$_{700}$ (**b**), the pathway productivity (**c**) and the pathway yield ratio (**d**) to key parameters. Asterisk indicates sensitivity or total sensitivity is significantly different ($P < 0.01$) from the dummy parameter (see Supplementary Note 4 for full analysis). **e**,**f**, OD$_{700}$ values (**e**) (calculated using Supplementary Table 9 for consistency) and resveratrol concentrations (**f**) of the seven pairs of co-cultures and the monoculture control (Mctrl) at 48 h in synthetic minimal medium. The co-culture setup and remaining p-coumaric acid concentrations in synthetic co-cultures are shown in Supplementary Fig. 32. $N = 3$ biologically independent samples and data are presented as mean ± s.d. Two-way ANOVA, followed by Turkey's multiple comparisons test with 95% confidence intervals were performed using GraphPad Prism 9.5.0 and $P$ values are noted.

(0.25 μM) at the 1:6 ratio (0.66 μM). In that ratio, C_Res1 accumulated 5.85 μM of the intermediate p-coumaric acid, which was not observed in Mctrl (Fig. 6e,f and Supplementary Fig. 32d).

Although co-culture C_Res1 improved resveratrol production, it is highly dependent on the initial population ratio. Cross-feeding behaviour increased resveratrol production under a wider range of

initial ratios in co-cultures via division of labour. Among the six pairs of cross-feeding co-cultures, AW_Res2 showed the best resveratrol production at all tested ratios; in particular, ratio 6:1 produced 0.79 µM resveratrol, 3.16-fold higher than production in Mctrl and 1.21-fold higher than in C_Res11:6. Interestingly, AW_Res2 also achieved a similarly high resveratrol production at initial ratio 20:1, which suggests the importance of the combination of differences in growth rates, metabolic and cross-feeding constraints in bioproduction. It was also observed that the order of sender and receiver strains affected both resveratrol production and growth in cross-feeding co-cultures; for example, AW_Res1 showed higher $OD_{700}$ values but lower resveratrol production than AW_Res2. In general, the inoculation ratios showed a clear trend with resveratrol production (Fig. 6e,f and Supplementary Fig. 32d).

Overall, division of labour can reduce metabolic burden and benefit resveratrol production in synthetic co-cultures. In addition, cross-feeding behaviour coupled to division of labour can further improve bioproduction and help maintain a more robust production under a wide range of initial ratios.

## Discussion

Here we designed and demonstrated the use of a toolkit for manipulating synthetic co-cultures of *S. cerevisiae*. Co-cultures are complex systems with multiple interactions between community members (for example, including metabolite exchange, different growth rates) and the wider environment (for example, substrate supplementation, secretion of products). We used an ensemble modelling approach to identify which interactions are key for determining co-culture dynamics, which showed that community dynamics in two- and three-member systems are controlled primarily by initial population ratios and exchange metabolite production rates. Our global sensitivity analysis inspired us to develop a toolkit for creating synthetic yeast co-cultures, composed of 15 auxotrophic strains and 15 target (essential) genes for metabolite overproduction (Supplementary Table 1). Using this toolkit, we created and characterized the growth dynamics of novel synthetic microbial communities including 60 pairs of two-member co-cultures, 5 pairs of three-member co-cultures via one-way communication and 4 pairs of three-member co-cultures via different two-way communication.

We tested four different approaches for controlling population growth rates, final population size and composition of co-cultures. These approaches comprised promoter engineering (governing metabolite exchange rates), different initial population ratios, different metabolite supplementations and different initial cell densities. As predicted by our ensemble modelling, these results showed that each approach was effective in controlling the growth and population size of the synthetic co-cultures. Engineering the strength of the promoters governing the expression of target genes, and therefore varying metabolite exchange, was shown to control the growth and population composition of five pairs of two-member co-cultures. Adjusting initial population ratios effectively altered growth and dynamics in co-cultures. Metabolite supplements also influenced co-culture behaviour, indicating a strategy for managing growth and composition. In addition, our experimental results show that initial cell density also influences population composition (as well as total growth), making it an alternative strategy to control growth and population ratio.

We demonstrated that synthetic co-cultures created with our toolkit can enhance production of metabolites of industrial interest. We selected the high-value antioxidant resveratrol as a case study due to its promise as a functional food, cosmetics ingredient and therapeutic[56]. The resveratrol pathway was split across either wild-type co-cultures (that is, no syntrophy) or the three most promising pairs of cross-feeding co-cultures: ade-Lys, Trp-ade and Trp-Lys. Engineering co-cultures and tuning population ratios can improve co-culture growth and optimize bioproduction.

In conclusion, here we report a modular toolkit for yeast co-culture construction governed by engineered cross-feeding. The kit consists of 15 auxotrophic strains (receiver cells) and 15 target genes for overproduction of essential metabolites (donor cells). Different co-cultures show distinct features (for example, different growth rates, population dynamics, final biomass), which can be used to guide their selection on the basis of the desired application (for example, mimicking behaviours observed on wild communities, or balancing biomass production to maximize product formation). We demonstrated four easily implemented strategies that can be used to control consortia growth, size and composition. Finally, we successfully applied our new toolkit to instantiate a metabolic division of labour system to produce a high-value aromatic compound.

## Methods

### Strains, media and chemicals

*Escherichia coli* Turbo Competent cells (NEB) were used for standard bacterial cloning and plasmid propagation. Selection and growth of *E. coli* was in Lysogeny Broth (LB) medium at 37 °C with aeration. Except when generating competent cells, the LB medium was supplemented with appropriate antibiotics (ampicillin 100 µg ml⁻¹, chloramphenicol 34 µg ml⁻¹ or kanamycin 50 µg ml⁻¹)[57].

Model yeast strain BY4741 (MATa *his3Δ1 leu2Δ0 met15Δ0 ura3Δ0*) was used as the wild-type strain in this study. Three culture media were used for yeast maintenance, including yeast extract peptone dextrose (YPD), synthetic complete dextrose (SD) and synthetic minimal medium (SM). YPD comprises of 10 g l⁻¹ yeast extract, 20 g l⁻¹ peptone and 20 g l⁻¹ glucose. SD comprises 6.7 g l⁻¹ yeast nitrogen base without amino acids; 1.4 g l⁻¹ yeast synthetic drop-out medium supplement without histidine, leucine, tryptophan and uracil; 20 g l⁻¹ glucose supplemented with histidine (20 mg l⁻¹), leucine (120 mg l⁻¹), tryptophan (20 mg l⁻¹) and uracil (20 mg l⁻¹) as necessary. SM is made up of 6.7 g l⁻¹ yeast nitrogen base without amino acids, 20 g l⁻¹ glucose, supplemented with amino acids following the protocol described in ref. 58. The yeast synthetic drop-out medium supplement for preparing SD was purchased from Sigma-Aldrich and from MP Biomedicals for preparing SM. Two percent bacteriological agar (VWR) was added when preparing plates. Yeast strains were stored in glycerol to a final concentration of 25% (v/v) at −80 °C.

All reagents, chemicals and analytical standards of amino acids, p-coumaric acid and resveratrol are listed as Supplementary Table 5.

### Plasmid construction and bacterial transformation

All plasmids in this study were created using the MoClo Yeast Toolkit (YTK) system[36] or the method described in ref. 57. Key gene information for the amino acid and nucleotide synthesis pathway is listed in Supplementary Table 1, and other parts or vector sequences in this study can be found either in the YTK system[36] or in ref. 57. All plasmid constructs used in this study are listed in Supplementary Table 7. Unless indicated, part sequences were either mutated or synthesized to remove or avoid all instances of BsmBI, BsaI, BpiI and NotI recognition sequences.

Golden Gate gene assembly was used to construct all plasmids in Supplementary Table 7. All parts were set to equimolar concentrations of 50 fmol ml⁻¹ (50 nM) before experiments. Golden Gate reactions were prepared as follows: 0.1 µl of backbone vector, 0.5 µl of each plasmid, 1 µl T4 DNA ligase buffer (Promega), 0.5 µl T7 DNA ligase (NEB), 0.5 µl restriction enzyme (BsaI or BsmBI; NEB) and water to bring the final volume to 10 µl. Reaction mixtures were then incubated in a thermocycler using the following programme: (42 °C for 2 min, 16 °C for 5 min) × 25 cycles, followed by a final digestion step at 60 °C for 10 min and then heat inactivation at 80 °C for 10 min. The entire reaction mixture was then ready for *E. coli* transformation, which was followed by a TSS (transformation storage solution) protocol for KCM (KCl, CaCl₂, MgCl₂) chemical transformation[59] before plating on LB plates with the appropriate antibiotics.

## Yeast transformation and colony PCR verification

Yeast transformation was performed using the lithium acetate protocol[60]. Chemically competent yeast cells were prepared as follows: fresh isolated colonies were cultured at 30 °C and 250 r.p.m. to saturation overnight in YPD. The following morning, the cells were diluted 1:100 in 10 ml fresh YPD in a 50 ml conical tube and incubated for 4–6 h to $OD_{600}$ 0.8–1.0 (measured using a spectrophotometer). Cells were pelleted and washed once with an equal volume of 0.1 M lithium acetate. Cells were then resuspended in 600 µl 0.1 M lithium acetate, and 100 µl of cells were aliquoted into individual 1.5 ml tubes and pelleted, ready for yeast transformation. Cells were resuspended in 64 µl of DNA/salmon sperm DNA mixture (10 µl of boiled salmon sperm DNA (10 mg ml$^{-1}$, Invitrogen) + (NotI digested) plasmids + double-distilled $H_2O$), then mixed with 294 µl of PEG/lithium acetate mixture (260 µl 50% (w/v) PEG-3350 + 36 µl 1 M lithium acetate). The yeast transformation mixture was then heat-shocked at 42 °C for 40 min, pelleted, resuspended in 200 µl 5 mM $CaCl_2$ and allowed to stand for 10 min before plating onto appropriate selection plates. Yeast colonies should come out after the plates were incubated at 30 °C for 2–3 days (or longer for some heavy burden or large genes).

Yeast transformation was verified by colony PCR using the Phire Plant Direct PCR master mix (F160L, Thermo Fisher). Isolated colonies (3–5) for each yeast transformation were selected and resuspended into 20–50 µl sterile water in PCR tubes. Each 10 µl PCR reaction system included 1 µl cell suspension, 5 µl 2X Phire Plant Direct PCR master mix, 0.5 µl forward primer, 0.5 µl reversed primer and 3 µl double-distilled $H_2O$. The PCR reactions were performed using a ProFlex PCR System (Thermo Fisher) under the recommended condition for Phire Plant polymerase: initial denaturation at 95 °C for 5 min, followed by 35 cycles of denaturation at 98 °C for 5 s, annealing at X °C for 5 s, extension at 72 °C at 20 s kb$^{-1}$, plus the final extension at 72 °C for 1 min (X represents the optimum annealing temperature for each primer pair). The 10 µl PCR reaction was then verified using agarose gel electrophoresis.

## Auxotrophic yeast construction

Auxotrophic yeasts were either taken from yeast knockout library from Markus Ralser's lab in the Francis Crick Institute, UK or constructed using the iterative markerless CRISPR-Cas9 genome editing method as described in the MoClo Yeast Toolkit (YTK)[36] and ref. [57]. For example, to generate auxotrophic strain BY4741 *arg4Δ*, a BpiI-digested Cas9 plasmid (pWS2081, URA+) was transformed into BY4741 along with BpiI-digested gRNA plasmids pHP071 and pHP072, and donor DNA. Two gRNA plasmids of pHP071 and pHP072 were generated by phosphorylating (standard T4 PNK reaction, NEB) and annealing primers oHP070 and oHP071, and oHP072 and oHP073, respectively, followed by a BsmBI Golden Gate reaction with SpCas9 gRNA gap repair vector pWS2069. Donor DNA was generated by PCR amplification of the *Arg4* region of BY4741 using primers oHP119 and oHP120, and a 20 bp landing pad (TAGCATGGTGACACAAGCAG) was used as a barcode in the donor DNA. Verification forward primer oHP147 and reversed primer oHP148 were designed at ~500 bp upstream and downstream of gene *Arg4*, respectively, and they were used to verify the *Arg4* deletion by colony PCR. The correct *Arg4* knockout strain should have ~1,000 bp size of PCR product. In addition, all auxotrophic strains were verified by colony PCR, Sanger sequencing and growth assay verification. Detailed information on primers, gRNAs, landing pads, donor DNA and knockout strains can be found in the list of oligos (Supplementary Table 6), plasmids (Supplementary Table 7) and strains (Supplementary Table 8).

## Monoculture and co-culture setup for microplate reader assay and bioproduction

### Seed culture and OD adjustment for co-culture setup.
Fresh isolated colonies of wild-type or verified engineered yeast strains were precultured in 2 ml of selective SC media at 30 °C, and 250 r.p.m. to saturation overnight. The following morning, 1 ml of preculture was taken and pelleted (3,000 × *g*, 1 min) in a 1.5 ml tube, then the cell pellet was washed three times (3,000 × *g*, 1 min) using SM medium and resuspended again in 1 ml SM medium. Washed cells (100 µl) were diluted 10–20 times before $OD_{600}$ measurement using cuvettes on a UV/Visible spectrophotometer (Biochrom WPA Lightwave II); the remaining 900 µl of washed cells were then pelleted and resuspended with SM medium to $OD_{600}$ 10 or $OD_{600}$ 20 measured using a spectrophotometer. The washed cells were then ready for the monoculture and co-culture setup described below.

**Monoculture and co-culture setup in microplate reader assay.** Monoculture was set up in a black 96-well plate (655090, Greiner Bio-One) by adding 5 µl $OD_{600}$ 20 (by spectrophotometer) individual washed cells and 120 µl SM medium (with/without 1.25 µl 100X or 2.5 µl 50X metabolite stock solution). The total monoculture volume was 125 µl with initial $OD_{600}$ 0.8 by spectrophotometer (equals $OD_{700}$ 0.102 by microplate reader). The monoculture with metabolite supplementation was used as positive control and the monoculture without metabolite supplementation was used as negative control. Similar to monocultures, all co-cultures used 125 µl culture volume in a black 96-well plate. In two-member (with different promoters) or three-member co-cultures with initial ratio 1:1 or 1:1:1, each washed member was loaded at 2.5 µl $OD_{600}$ 20 (by spectrophotometer) into SM medium with initial $OD_{600}$ 0.8 or 1.2 by spectrophotometer (equals $OD_{700}$ 0.102 or 0.125 by microplate reader) in two-member or three-member co-cultures. In two-member co-cultures with different initial ratios, we adjusted the cell dosage volume of each member to match the ratios 10:1, 1:1 and 1:10. In co-cultures with different metabolite supplementations, the dosages (mg l$^{-1}$) were as follows: adenine at 2.5, 5, 10; lysine at 12.5, 25, 50; histidine at 10, 20, 40; and tryptophan at 10, 20, 40. In three-member co-cultures with different initial cell densities, the initial $OD_{600}$ value (by spectrophotometer) for each member was 0.2, 0.4, 0.8 and 1.6, respectively (equals $OD_{700}$ 0.067, 0.078, 0.102 and 0.148 by microplate reader). The SPARK multimode microplate reader (Tecan) was used for recording the $OD_{700}$ values and fluorescence intensities of RFP, BFP and GFP in monoculture and co-cultures. The standard curves for $OD_{700}$ using the microplate reader and $OD_{600}$ using the spectrophotometer can be found in Supplementary Table 9.

**Monoculture and two-member co-culture for resveratrol production.** Monocultures and co-cultures were performed using deep 96-well plates in 500 µl volume for resveratrol production. Monocultures were used as negative controls. In monocultures, the initial $OD_{600}$ value was set at 0.8 (by spectrophotometer, equals $OD_{700}$ 0.102 by microplate reader) for each strain, and the 500 µl volume included 40 µl of $OD_{600}$ 10 (by spectrophotometer) individual washed seed culture plus 460 µl SM medium. In co-cultures, the initial total $OD_{600}$ was set as 0.8 (by spectrophotometer, equals $OD_{700}$ 0.102 by microplate reader), and the 500 µl volume included 40 µl of $OD_{600}$ 10 mixed washed two members plus 460 µl SM medium. The two members were inoculated at different initial ratios of 20:1, 6:1, 1:1, 1:6 and 1:20. The deep 96-well plates were incubated at 30 °C and 250 r.p.m. for 72 h using InforsHT Multitron incubators.

## OD measurement, plate reader assay and flow cytometry analysis

The endpoint $OD_{600}$ values of seed cultures in tubes were measured using cuvettes in a UV/Visible spectrophotometer (Biochrom WPA Lightwave II) after 10–20 times dilution. The endpoint $OD_{700}$ values of cultures in deep 96-well plates were measured using Magellan Standard software for a SPARK multimode microplate reader (Tecan). To make the OD values easy to compare, two standard curves were prepared to convert both $OD_{600}$ values from spectrophotometer and microplate reader into $OD_{700}$ scale by microplate reader (Supplementary Table 9). Unless explicitly indicated, all $OD_{600}$ values shown are from

spectrophotometer readings, and all $OD_{700}$ values shown are from or were converted to the Tecan microplate reader scale. Morevoer, this SPARK multimode microplate reader (Tecan) was used for setting up the kinetic cell cultures in a black 96-well plate (655090, Greiner Bio-One) at 30 °C with 270 r.p.m. double orbital continuous shaking for 48 h or 72 h. It recorded $OD_{700}$ values and different fluorescence intensities including for mScarlet-I, mTagBFP2 and sfGFP (abbreviated as RFP, BFP and GFP). The excitation and emission wavelengths (nm) for RFP, BFP and GFP were set at 560/620, 400/465 and 485/535, respectively. An Attune NxT flow cytometer v.3.1 (Thermo Scientific) was used for analysis of the population percentages of subpopulations in yeast co-cultures. The cytometer setting for measuring the above RFP, BFP and GFP was as follows: FSC 130 V, SSC 340 V, BL1 410 V, VL1 370 V and YL2 530 V. Fluorescence data were collected from >10,000 cells for each sample and analysed using FlowJo v.10.8.1 software (BD Biosciences). The detailed gating strategy for these flow cytometer data is shown as Supplementary Fig. 33.

## LC−MS quantification of metabolites in co-cultures

Cell cultures (500 µl) were centrifuged at 2,500g for 5 min to pellet the cells. The growth medium (100 µl) was transferred to a second centrifuge tube, mixed with 400 µl 50% acetonitrile and centrifuged at 10,000g for 5 min. Then, 1 µl of the supernatant was subjected to LC−MS analysis. An Agilent 1290 Infinity system was used to analyse these prepared samples in combination with an Agilent 6550 quadrupole time-of-flight (Q-ToF) mass spectrometer. Chromatographic separation was performed on an Agilent Poroshell 120 HILIC-Z column (2.1 mm × 100 mm, 1.9 µm, p/n 685675-924) at a temperature of 30 °C using two different solvent systems. Buffer A was 10 mM ammonium formate in water and buffer B was 10 mM ammonium formate in water/acetonitrile (10:90 v:v). Starting at 100% buffer B, LC was performed at a solvent flow rate of 0.25 ml min⁻¹ with a linear gradient to 70% buffer B over 11.5 min, with a further decrease to 60% B over 1 min. Injection volume was 1 µl and negative ion spectra were recorded between a mass range of 100−1,000 $m/z$ at a rate of 1 spectrum per second. The prepared calibration curves of standards included glucose, various amino acids and nucleotides. Quantitation was based on the MS peak area of precursor or fragment ions in comparison with the analytical standards. Positive ion detection mode was used for amino acids, nucleotides and glucose samples. The results were analysed using Agilent MassHunter Qualitative Analysis v.10. Error bars represent standard deviations from two independent biological samples.

## LC−MS analysis of metabolites in the resveratrol synthesis pathway

Cell culture (300 µl) was mixed with an equal volume of ethanol by incubating at 700 r.p.m. at 30 °C for 5 min, then centrifuging at 2,500g for 30 min before loading the supernatants into a 96-well sample plate for LC−MS analysis as previously described[61,62]. An Agilent 1290 Infinity system was used to analyse these prepared samples with an online diode array detector in combination with an Agilent 6500 Q-ToF mass spectrometer. An Agilent Eclipse Plus C18 2.1 × 50 mm (1.8 µm particle size) column was used at a temperature of 25 °C, with a solvent flow rate of 0.2 ml min⁻¹. LC was performed with a linear gradient of buffer A (0.1% formic acid) and buffer B (0.1% formic acid in acetonitrile) from 2% to 98% buffer B over 2.5 min, which was held at 98% buffer B for 1 min. Injection volume was 1 µl and spectra were recorded between a mass range of 90−1,000 $m/z$ at a rate of 3 spectra per second. The prepared calibration curves of standards included p-coumaric acid and resveratrol. Quantitation was based on the MS peak area of precursor or fragment ions in comparison with the analytical standards. Negative ion detection mode was used for resveratrol samples. Error bars represent standard deviations from three independent biological samples.

## Mathematical model of the co-culture system

To identify the key design parameters of the system, a chemostat modelling framework for co-cultures and microbial cross-feeding recently developed was adopted[20]. The framework was updated for batch culture growth and production or utilization of multiple amino acids per strain. The modelling framework consists of a series of coupled ordinary differential equations that capture the time evolution of the extracellular glucose ($G$), metabolites ($x_i$) and the population of each strain ($y_i$). The strain that has been engineered to overproduce metabolite $x_j$ is denoted $y_j$. This strain produces metabolite $x_j$ and is auxotrophic for all other amino acids $x_i$ where $i \neq j$.

All strains take up glucose at rate $J^{y_i}_{upt,G}$, where $y_i$ denotes the strain. Therefore, the dynamics of the glucose concentration are:

$$\frac{dG}{dt} = -\sum \left( J^{y_i}_{upt,G} y_i \right) \tag{1}$$

Strains grow and die (decay) at rates $J^{y_i}_{grow}$ and $\eta_{y_i}$, respectively, giving the dynamics of the strain population as:

$$\frac{dy_i}{dt} = \left( J^{y_i}_{grow} - \eta_{y_i} \right) y_i \tag{2}$$

The exchange metabolite $x_i$ is produced at rate $J^{y_i}_{leak,i}$ by strain $i$ and consumed at rate $J^{y_j}_{upt,i}$ by auxotrophic strains (denoted in this case, $j$ is the set of strains that consume metabolite $x_i$). The dynamics of the metabolite $x_i$ are given by:

$$\frac{dx_i}{dt} = J^{y_i}_{leak,i} y_i - \sum_{j \neq i} \left( J^{y_j}_{upt,i} y_j \right) \tag{3}$$

The uptake rates of glucose and exchange metabolites were modelled using Monod kinetics, where the maximum uptake rate and the Michaelis constants are denoted as $V_{max,G}$ and $k_{M,G}$ for glucose and $V_{max,i}$ and $k_{M,i}$ for exchange metabolite $i$:

$$J^{y_i}_{upt,G} = \frac{V_{max,G}\, G}{k_{M,G} + G} \text{ and } J^{y_i}_{upt,j} = \frac{V_{max,j}\, x_j}{k_{M,j} + x_j} \tag{4}$$

The exchanged metabolite production flux was assumed to be proportional to the glucose flux, such that:

$$J^{y_i}_{leak,i} = \phi_i\, \delta_i\, J^{y_i}_{upt,G} \tag{5}$$

where $\phi_i$ governs the proportion of the glucose flux that is diverted to exchange metabolite $x_i$ biosynthesis and $\delta_i$ is the number of glucose molecules required to produce an amino acid. $\delta_1$ is set to 1 throughout unless otherwise stated. A thorough discussion of this derivation is provided in ref. 20.

Assuming that the strain's growth is limited by glucose or the exchange metabolites that the strain is auxotrophic for (in this case $x_j$), the growth rate can be calculated as:

$$J^{y_i}_{grow} = \min \left( J^{y_i}_{grow,G}, J^{y_i}_{grow,j} \right) \tag{6}$$

The growth rate on the exchange metabolite $x_j$ was assumed to be proportional to its uptake flux:

$$J^{y_i}_{grow,j} = \gamma_j J^{y_i}_{upt,j} \tag{7}$$

with a constant of proportionality (that is, biomass yield) of $\gamma_j$.

The growth on glucose was assumed to be proportional to the glucose flux not utilized to make the exchange metabolite $x_i$, that is, proportional to $(1-\phi_i)$, with a constant of proportionality of $\gamma_G$:

$$J^{y_i}_{grow,G} = \gamma_G (1 - \phi_i) J^{y_i}_{upt,G} \tag{8}$$

Updates to the model to account for competition between more than two strains, toxicity of metabolites and the production of the heterologous metabolic pathways are described in Supplementary Notes 1–4.

## Extended Fourier amplitude global sensitivity analysis

Biologically permissible ranges for each parameter were obtained through a combination of literature search and initial experimentation to derive a nominal parameter set. This nominal parameterization showed good agreement with the single-strain growth curves for the used population over time, as measured by $OD_{700}$. To untangle how each parameter contributed to the behaviour of the system, a global sensitivity analysis approach developed previously[63] was used. In brief, this method is based on the extended Fourier amplitude sensitivity test (eFAST), which works by systematically varying model inputs (parameters). A predefined sinusoidal function is used to ensure that the whole parameter space is searched and no region is oversampled. The model is simulated for each input and its behaviour is captured as predefined output metrics (here, the final population, the batch culture time, the final population ratios as well as the maximum growth rate of each strain, the maximum uptake rate of each amino acid by its respective auxotroph and the production rate of each amino acid). The parameter sampling method was modified such that the initial ratio of the strains sums to one. The model was sampled for multiple runs. This creates a 'noisy' trace with model output varying over each run number. The algorithm then utilizes the Fourier transform to extract the variance at each frequency. Each frequency (and its harmonics) corresponds to an input parameter (as determined by the predefined sinusoidal function). The first-order sensitivity, the direct impact of a parameter on the model output, is the sum of the variance at the known frequency and its harmonics. The total-order effect/sensitivity is the total sum of the variance (across all frequencies), which captures the impact of the interactions the given parameter may have with any other parameters. To enable more efficient parameter sampling, the eFAST method randomly resamples the parameter search curves. While this increases computational efficiency, it can introduce small but non-zero sensitivity indices for parameters to which the model has no sensitivity to. To enable identification of this effect, a 'dummy parameter' was deliberately introduced into the analysis. This parameter is varied in the global sensitivity analysis but does not contribute to the model dynamics, that is, it does not appear in the model equations; however, the sensitivity analysis will produce sensitivity indices for this dummy parameter. As previously described[63], a two-sample $t$-test was used to identify where the mean index from the resample procedure is significantly different from that produced for the dummy parameter which has no impact on the model. The sensitivity analysis was run using 100 resamplings with 1,285 samples per search and 4 Fourier coefficients retained. A $P$-value significance threshold of 1% was chosen but updated using Bonferroni correction to account for multiple testing. The significance threshold for each analysis is therefore $0.01/n_k$ where $n_k$ is the number of parameters varied in that analysis. The specific parameters varied in each sensitivity analysis are reported in the respective figures and full results are shown in the Supplementary figures. Model parameters were varied on a linear uniform scale as follows: $N_0 = [0.01\ldots1]$ ($OD_{700}$), $r_{0,i} = [0.01\ldots1]$ (unitless ratio), $x_{0,i} = [0\ldots75]$ (mg l$^{-1}$), $\gamma_G = \gamma_i = [0.01\ldots1]$ (biomass yield per g or mg), $V_{max,G} = [1\ldots30]$ (g h$^{-1}$), $K_{M,G} = [1\ldots100]$ (g), $V_{max,i} = [1\ldots120]$ (mg h$^{-1}$), $K_{M,i} = [1\ldots1000]$ (mg) and $\phi_i = [0.01\ldots0.5]$ (unitless ratio).

## Statistical analysis and reproducibility

All mathematical simulations and related statistical analysis were carried out in MATLAB 2019a or MATLAB 2021a (Mathworks) using the in-built stiff solver ode15s. Unless explicitly indicated, all wet-lab experiment data were subjected to analysis using Microsoft Excel 365 and Prism 9.5.0 (GraphPad) software. The error bars or bands

presented in the figures correspond to the standard deviation, as specified in figure legends. Statistical analyses were conducted using either one-way or two-way analysis of variance (ANOVA), followed by Turkey's or Bonferroni's multiple comparisons test with 95% confidence intervals, and $P$ values are noted.

## Reporting summary

Further information on research design is available in the Nature Portfolio Reporting Summary linked to this article.

## Data availability

All source data are publicly available, provided as (supplementary) source data, Supplementary Tables 1–9 or published in GitHub. Inkscape v.1.2 software was used to draw diagrams and assemble figures. Raw flow cytometry data for Figs. 4 and 5 are available at https://github.com/hdpeng89/Raw-flow-cytometry-data-yeast-co-culture. Source data are provided with this paper.

## Code availability

The MATLAB source code (with exemplar analysis runs as .mat files) is available on the Zenodo repository[64]. Each .mat file contains the sampling results as raw data and the results of our processing and statistical tests.

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

## Acknowledgements

R.L.-A. received funding from BBSRC (BB/R01602X/1, BB/T013176/1, BB/T011408/1 - 19-ERACoBioTech-33 SyCoLim), British Council 527429894, Newton Advanced Fellowship (NAF\R1\201187), Yeast4Bio Cost Action 18229, European Research Council (ERC) (DEUSBIO - 949080) and the Bio-based Industries Joint (PERFECOAT - 101022370) under the European Union's Horizon 2020 research and innovation programme. H.P. received funding from the European Union's Horizon 2020 research and innovation programme under Marie Skłodowska-Curie grant agreement No. 899987. A.P.S.D. received funding from the Royal Academy of Engineering under their Research Fellowship Scheme. We thank the support for the yeast modular cloning toolkit (YTK) from the W. Shaw and T. Ellis labs; D. J. Bell for the analytical support from the SynbiCITE Innovation and Knowledge Centre at Imperial College London; M. Mülleder and M. Ralser labs for providing the yeast Prototrophic toolkit pHLUM v.2.

## Author contributions

R.L.-A. and H.P. conceptualized the project and designed experiments; H.P. conducted the wet-lab experiments, data analysis and visualization, and wrote the original draft; A.P.S.D. conducted the mathematical modelling analysis, visualization and writing; E.J.S. revised and polished the manuscript and visualization; H.-H.C. and W.J. conducted some experiments; R.L.-A. revised and polished the manuscript, supervised and administered the project, and acquired funding. All authors reviewed and provided input on the manuscript.

## Competing interests

The authors declare no competing interests.

## Additional information

**Extended data** is available for this paper at https://doi.org/10.1038/s41564-023-01596-4.

**Correspondence and requests for materials** should be addressed to Rodrigo Ledesma-Amaro.

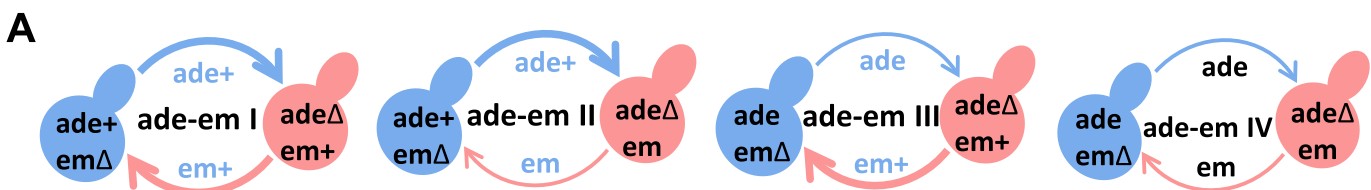

**A**

*em include arg, cys, his, leu, lys, met, phe, tyr, ser, thr, trp, ura, val, ile

**B**

**C**

OD$_{700nm}$ ROD$_{700nm}$ BOD$_{700nm}$

**D**

**Extended Data Fig. 1 | See next page for caption.**

**Extended Data Fig. 1 | Target screening for co-culture potential using 52 pairs of adenine-exchanged metabolite cross-feeding co-cultures. a**. 52 pairs of adenine-exchanged metabolite (ade-em) two-member cross-feeding co-cultures were created for each target, including ade-em I, overexpression of em and ade in each member; ade-em II, overexpression of ade only in blue; ade-em III, overexpression of em only in red; ade-em IV, no overexpression in each member. Em includes arg, cys, his, leu, lys, met, phe, tyr, ser, thr, trp, ura, val, ile. BY4741-pHLUM was used as the positive control (+ve). These co-cultures were tested with the initial ratio of 1:1 and initial cell density of each member was $OD_{700nm}$ 0.078 by a Tecan Spark plate reader. **b**. $OD_{700nm}$ values of negative controls for monocultures at 72 h. The strain adeΔem (red) was duplicated in the

diagram to correspond with the numbers of co-culture combinations. **c**. Estimated maximum $OD_{700nm}$ values of red population (ROD), blue population (BOD) and total OD values (ROD + BOD) in ade-em co-cultures within 72 h. The standard curves of estimating the fluorescent intensities of BFP and RFP to OD values were shown in Supplementary Table 3. **d**. The maximal measured total $OD_{700nm}$ values of 52 pairs of adenine-exchanged metabolite co-cultures at 72 h are close to the sum of estimated values of BOD and ROD. N = three biologically independent samples, and data are presented as mean values +/− SD. Two-way ANOVA, followed by Bonferroni's multiple comparisons test with 95% confidence intervals were performed using Prism 9.5.0 (GraphPad) software, and p values were noted.

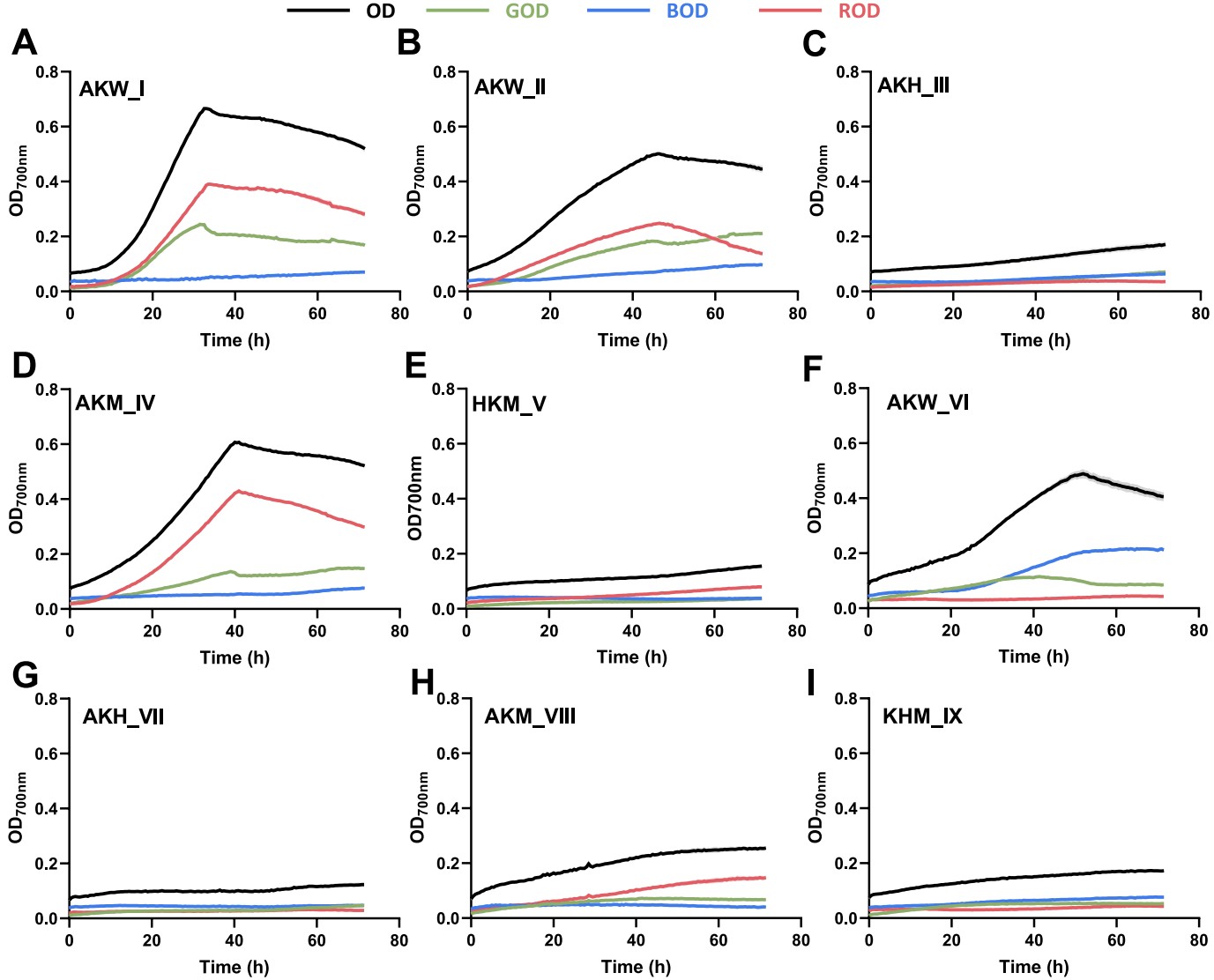

**Extended Data Fig. 2 | Time courses of cell growth of nine three-member co-cultures and their respective individual members. a-i**. Three-member co-cultures include AKW_I, AKW_II, AKH_III, AKM_IV, HKM_V, AKW_VI, AKH_VII, AKM_VIII, KHM_IX. The initial cell density for each member was $OD_{700nm}$ 0.078, and the initial ratio of each member was 1:1:1 for these co-cultures. GOD, BOD and ROD are estimated cell density tagged with GFP, BFP and RFP respectively, which were calculated using the standard curves between OD values and fluorescence intensities (GFP, BFP and RFP). N = three biologically independent samples, and data are presented as mean values +/− SD.

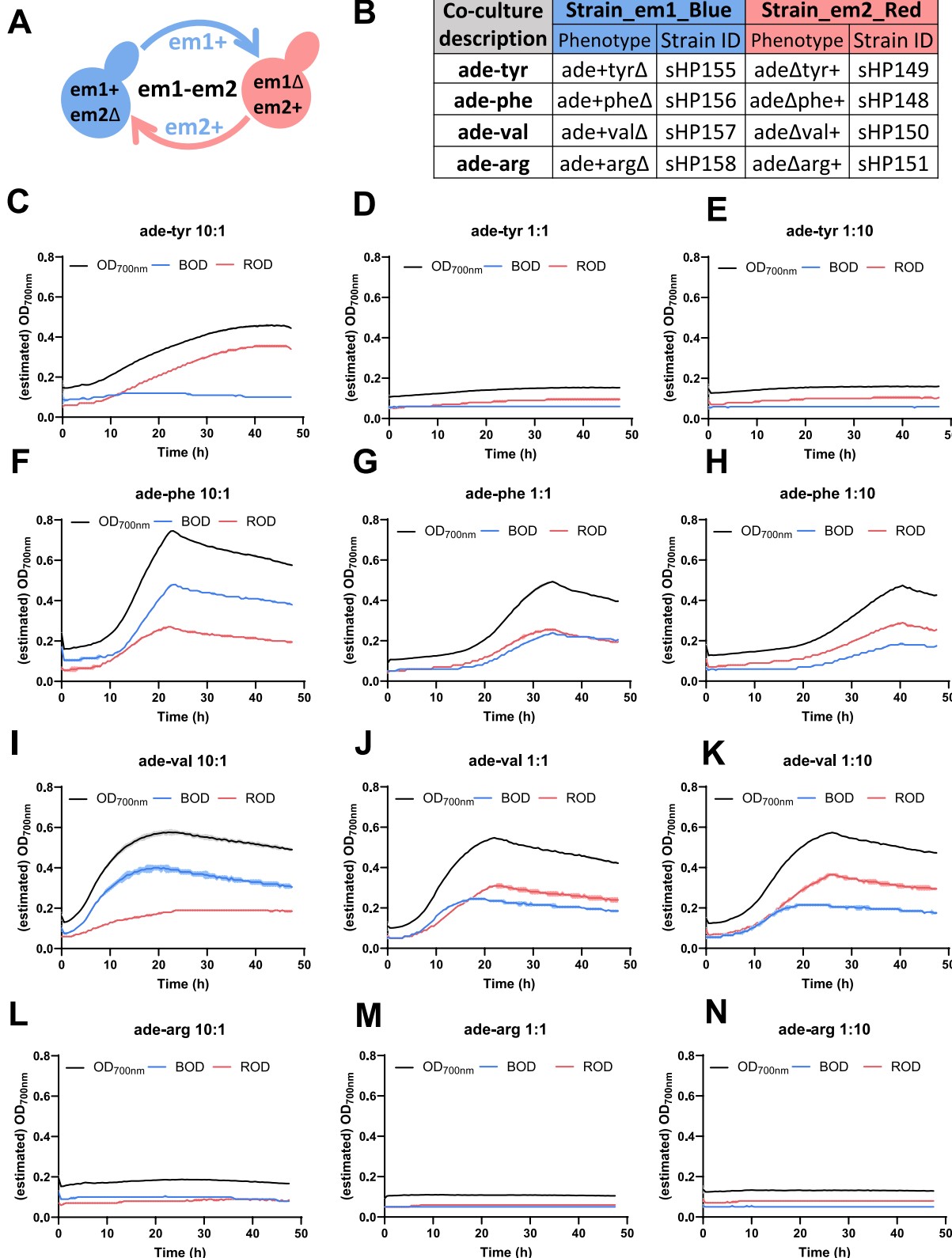

**Extended Data Fig. 3 | Different initial ratios adjusted cell growth and population size of four pairs of two-member cross-feeding co-cultures.**
**a**. Diagram of two-member cross-feeding co-cultures em1-em2. **b**. Strain table of four pairs of co-cultures including ade-tyr, ade-phe, ade-val, and ade-arg.
**c-n**. Time courses of co-culture growth (OD$_{700nm}$) and estimated growth of red and blue members (ROD and BOD) in 48 h. Four pairs of two-member cross-feeding co-cultures include ade-tyr, ade-phe, ade-val, ade-arg, which were tested in three different initial ratios of 10:1, 1:1 and 1:10, respectively. In the co-culture setup, the initial cell density for each member was OD$_{700nm}$ 0.078, and the initial ratios included 10:1, 1:1 and 1:10. The cell growth and fluorescent intensities (red and blue) were monitored by Tecan Spark plate reader for 48 h in the synthetic minimal medium. In the co-culture ade-val, isoleucine was supplemented in the synthetic minimal medium. N = three biologically independent samples, and data are presented as mean values +/− SD.

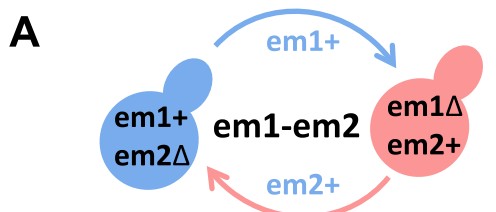

**A**

| Co-culture | Strain_em1_Blue | | | Strain_em2_Red | | |
|---|---|---|---|---|---|---|
| description | Phenotype | No. | Strain ID | Phenotype | No. | Strain ID |
| **ade#3-lys#2** | ade+lysΔ | ade#3 | sHP063 | adeΔlys+ | lys#2 | sHP067 |
| **his#1-lys#2** | hisΔlys+ | his#1 | sHP266 | his+lysΔ | lys#2 | sHP272 |
| **trp#1-leu#4** | trp+leuΔ | trp#4 | sHP099 | trpΔleu+ | leu#1 | sHP102 |

**B**

**C**

RFP%
BFP%

**Extended Data Fig. 4 | Effects of metabolite supplementation on growth and population percentages of three pairs of two-member co-cultures.**
**a.** Diagram of two-member cross-feeding co-culture and strain combination table; **b.** Maximal cell growth of three pairs of co-cultures within 72 h with/without metabolite supplementation; **c.** Population percentages of each member of three pairs of co-cultures at 72 h with/without metabolite supplementation.

Metabolite dosages mg/L were ade 2.5, 5, 10; lys 12.5, 25, 50; his 5, 10, 20; 12.5, 25, 50; trp 10, 20, 40; leu 15, 30, 60, respectively. Ade_2.5 means the final medium contains 2.5 mg/L ade supplementation. N = three biologically independent samples, and data are presented as mean values +/− SD. One-way ANOVA, followed by Bonferroni's multiple comparisons test with 95% confidence intervals were performed using Prism 9.5.0 (GraphPad) software, and p values were noted.

# Reporting Summary

## Statistics

For all statistical analyses, confirm that the following items are present in the figure legend, table legend, main text, or Methods section.

| n/a | Confirmed | |
|---|---|---|
| ☐ | ☒ | The exact sample size (*n*) for each experimental group/condition, given as a discrete number and unit of measurement |
| ☐ | ☒ | A statement on whether measurements were taken from distinct samples or whether the same sample was measured repeatedly |
| ☐ | ☒ | The statistical test(s) used AND whether they are one- or two-sided<br>*Only common tests should be described solely by name; describe more complex techniques in the Methods section.* |
| ☒ | ☐ | A description of all covariates tested |
| ☐ | ☒ | A description of any assumptions or corrections, such as tests of normality and adjustment for multiple comparisons |
| ☐ | ☒ | A full description of the statistical parameters including central tendency (e.g. means) or other basic estimates (e.g. regression coefficient) AND variation (e.g. standard deviation) or associated estimates of uncertainty (e.g. confidence intervals) |
| ☐ | ☒ | For null hypothesis testing, the test statistic (e.g. $F$, $t$, $r$) with confidence intervals, effect sizes, degrees of freedom and $P$ value noted<br>*Give P values as exact values whenever suitable.* |
| ☒ | ☐ | For Bayesian analysis, information on the choice of priors and Markov chain Monte Carlo settings |
| ☒ | ☐ | For hierarchical and complex designs, identification of the appropriate level for tests and full reporting of outcomes |
| ☒ | ☐ | Estimates of effect sizes (e.g. Cohen's *d*, Pearson's *r*), indicating how they were calculated |

*Our web collection on statistics for biologists contains articles on many of the points above.*

## Software and code

Policy information about availability of computer code

| | |
|---|---|
| Data collection | 1. Magellan Standard was used to collect data from the Tecan Spark plate reader.<br>2. Attune NxT software v3.1 was used to collect flow cytometry data.<br>3. Agilent MassHunt version 10 was used to collect LC-MS data. |
| Data analysis | 1. GraphPad Prism 9 was used for generating most graphs and statistical analysis for wet lab experiments.<br>2. FlowJo v10.6.2 was used to analyse flow cytometry data.<br>3. Microsoft Excel was used to calculate the changes of cell growth OD values, fluorescent intensities, metabolite concentrations.<br>4. Benchling was used for designing all nucleotide sequences and CRISPR experiments.<br>5. MassHunter Quantitative software version 10 was used to analyse LC-MS data.<br>6. All simulations and related statistical analysis were carried out in MATLAB 2019a or MATLAB 2021a (both Mathworks Inc, MA, USA) using the in-built stiff solver ode15s. The global sensitivity toolbox, implemented in MATLAB, developed by Marino et al. was retrieved from http://malthus.micro.med.umich.edu/lab/usadata/ and used as detailed in their original publication.<br>7. Inkscape v1.2 was used to draw diagrams and assemble figures. |

For manuscripts utilizing custom algorithms or software that are central to the research but not yet described in published literature, software must be made available to editors and reviewers. We strongly encourage code deposition in a community repository (e.g. GitHub). See the Nature Portfolio guidelines for submitting code & software for further information.

## Data

Policy information about availability of data

All manuscripts must include a data availability statement. This statement should provide the following information, where applicable:

- Accession codes, unique identifiers, or web links for publicly available datasets
- A description of any restrictions on data availability
- For clinical datasets or third party data, please ensure that the statement adheres to our policy

-All source data are publicly available, either provided as (supplementary) source data with this paper, or published in public repository. Raw flow cytometry data for Fig. 4 & 5 are available on https://github.com/hdpeng89/Raw-flow-cytometry-data-yeast-co-culture.
-The MATLAB source code (with exemplar analysis runs as .mat files) was provided (with exemplar analysis runs as .mat files) on the Zendoo repository (DOI: 10.5281/zenodo.10257825).

## Research involving human participants, their data, or biological material

Policy information about studies with human participants or human data. See also policy information about sex, gender (identity/presentation), and sexual orientation and race, ethnicity and racism.

| Reporting on sex and gender | N/A |
| --- | --- |
| Reporting on race, ethnicity, or other socially relevant groupings | N/A |
| Population characteristics | N/A |
| Recruitment | N/A |
| Ethics oversight | N/A |

Note that full information on the approval of the study protocol must also be provided in the manuscript.

# Field-specific reporting

Please select the one below that is the best fit for your research. If you are not sure, read the appropriate sections before making your selection.

☒ Life sciences          ☐ Behavioural & social sciences          ☐ Ecological, evolutionary & environmental sciences

For a reference copy of the document with all sections, see nature.com/documents/nr-reporting-summary-flat.pdf

# Life sciences study design

All studies must disclose on these points even when the disclosure is negative.

| Sample size | No sample size calculation was performed, experiments were performed in triplicates (n = 3) or greater, which is the generally accepted by the scientific community. |
| --- | --- |
| Data exclusions | No data were excluded from the manuscript. |
| Replication | All experiments were performed in triplicates or greater and all attempts at replication were successful (sample size indicated in figure legend). |
| Randomization | Transformed yeast colonies were chosen at random from plates and no data was excluded. |
| Blinding | The study does not contain experiments where blinding would be applicable. |

# Reporting for specific materials, systems and methods

We require information from authors about some types of materials, experimental systems and methods used in many studies. Here, indicate whether each material, system or method listed is relevant to your study. If you are not sure if a list item applies to your research, read the appropriate section before selecting a response.

## Materials & experimental systems

| n/a | Involved in the study |
|---|---|
| ☒ | ☐ Antibodies |
| ☐ | ☒ Eukaryotic cell lines |
| ☒ | ☐ Palaeontology and archaeology |
| ☒ | ☐ Animals and other organisms |
| ☒ | ☐ Clinical data |
| ☒ | ☐ Dual use research of concern |
| ☒ | ☐ Plants |

## Methods

| n/a | Involved in the study |
|---|---|
| ☒ | ☐ ChIP-seq |
| ☐ | ☒ Flow cytometry |
| ☒ | ☐ MRI-based neuroimaging |

# Eukaryotic cell lines

Policy information about cell lines and Sex and Gender in Research

| | |
|---|---|
| Cell line source(s) | Saccharomyces cerevisiae Strain BY4741 from ATCC. |
| Authentication | We confirmed all derivative strains by colony PCR and sequencing. |
| Mycoplasma contamination | Yeast does not have this contamination. |
| Commonly misidentified lines (See ICLAC register) | No common misidentified lines were used. |

# Flow Cytometry

## Plots

Confirm that:

☒ The axis labels state the marker and fluorochrome used (e.g. CD4-FITC).

☒ The axis scales are clearly visible. Include numbers along axes only for bottom left plot of group (a 'group' is an analysis of identical markers).

☒ All plots are contour plots with outliers or pseudocolor plots.

☒ A numerical value for number of cells or percentage (with statistics) is provided.

## Methodology

| | |
|---|---|
| Sample preparation | Cell cultures were diluted 4-5 dilutions using the culture synthetic minimal medium before measurement. |
| Instrument | Attune NxT3 Colour with Autosampler |
| Software | Attune NxT software for collection. FlowJo version 10.6 for data analysis. |
| Cell population abundance | Typical samples included at least 10,000 cells. |
| Gating strategy | Yeast cells were gated fro singlets using FSC-H vs FSC-A and to remove background noicse. No other Gatings was performed on global yeast population. > 10, 000 events were collected and analysed within the singlets gate fro each measurement. |

☒ Tick this box to confirm that a figure exemplifying the gating strategy is provided in the Supplementary Information.

