## [Peer Review File · Nature Microbiology]

Peer Review Information

Journal: Nature Microbiology

Manuscript Title: A molecular toolkit of cross-feeding strains for engineering synthetic yeast communities

Corresponding author name(s): Dr Rodrigo Ledesma-Amaro

Reviewer Comments & Decisions:Decision Letter, initial version:

Message: 3rd July 2023

Dear Professor Ledesma-Amaro,

Thank you for your patience while your manuscript "Engineering synthetic yeast communities with a molecular cross-feeding toolkit" was under peer-review at Nature Microbiology. It has now been seen by 3 referees, whose expertise and comments you will find at the end of this email. Although they find your work of some potential interest, they have raised a number of concerns that will need to be addressed before we can consider publication of the work in Nature Microbiology.

In particular, the referees ask to better explain the model, to more carefully use statistical analyses, and to perform some additional experiments (as requested by referee #2). Referee #3 also has some concerns regarding the model. The referee concerns should be addressed in full.

Should further experimental data allow you to address these criticisms, we would be happy to look at a revised manuscript.

Please include a data availability statement as a separate section after Methods but before references, under the heading "Data Availability". This section should inform readers about the availability of the data used to support the conclusions of your study. This information includes accession codes to public repositories (data banks for protein, DNA or RNA sequences, microarray, proteomics data etc...), references to source data published alongside the paper, unique identifiers such as URLs to data repository entries, or data set DOIs, and any other statement about data availability. At a minimum, you should include the following statement: "The data that support the findings of this study are available from the corresponding author upon request", mentioning any restrictions on availability. If DOIs are provided, we also strongly encourage including these in the Reference list (authors, title, publisher (repository name), identifier, year). For more guidance on how to write this section please see: <http://www.nature.com/authors/policies/data/data-availability-statements-data-citations.pdf>

* If you have not done so already we suggest that you begin to revise your manuscript so that it conforms to our Resource format instructions at <http://www.nature.com/nmicrobiol/info/final-submission>. Refer also to any guidelines provided in this letter.

When submitting the revised version of your manuscript, please pay close attention to our [href="https://www.nature.com/nature-portfolio/editorial-policies/image-integrity">Digital Image Integrity Guidelines.](https://www.nature.com/nature-portfolio/editorial-policies/image-integrity) and to the following points below:

Note: This url links to your confidential homepage and associated information about manuscripts you may have submitted or be reviewing for us. If you wish to forward this e-mail to co-authors, please delete this link to your homepage first.

Nature Microbiology is committed to improving transparency in authorship. As part of our efforts in this direction, we are now requesting that all authors identified as 'corresponding author' on published papers create and link their Open Researcher and Contributor Identifier (ORCID) with their account on the Manuscript Tracking System (MTS), prior to acceptance. This applies to primary research papers only. ORCID helps the scientific community achieve unambiguous attribution of all scholarly contributions. You can create and link your ORCID from the home page of the MTS by clicking on 'Modify my Springer Nature account'. For more information please visit www.springernature.com/orcid.

If you wish to submit a suitably revised manuscript we would hope to receive it within 6 months. If you cannot send it within this time, please let us know. We will be happy to consider your revision, even if a similar study has been accepted for publication at Nature Microbiology or published elsewhere (up to a maximum of 4 months).

Yours sincerely,

Reviewer Expertise:

Referee #1: Synthetic biology, synthetic microbial communities

Referee #2: Synthetic communities, yeast

Referee #3: Modelling, synthetic biology

Reviewer Comments:

Reviewer #1 (Remarks to the Author):

The authors have developed a toolkit for creating co-cultures of *Saccharomyces cerevisiae*, an important industrial organism. They engineered yeast strains that can exchange essential metabolites, allowing the formation of multi-strain consortia. Through modelling and experiments, they investigate the factors influencing population dynamics in these co-culture systems. The toolkit is demonstrated to enhance and tune resveratrol production. Overall, this work provides a valuable resource for synthetic ecology and biomanufacturing applications.

Major comments:

- **Whole manuscript:** To ensure consistency and clarity throughout the manuscript, I suggest that the authors specify which wavelength of optical density (OD) measurement was used whenever referring to OD values. Since two different ODs are presented in the manuscript (600nm and 700nm), including the wavelength information alongside OD measurements would help avoid confusion and provide a clear understanding of the reported results. Furthermore, to adhere to the standard convention for writing optical density (OD), I recommend revising the manuscript to include the subscript denoting the wavelength whenever OD is mentioned.
- **Method section:** To ensure clarity and consistency, I recommend that the authors specify the statistical methods used in the Methods section, including the specific tests employed (e.g., t-tests, ANOVA) and the corresponding software used for the statistical analysis. Additionally, it would be beneficial to include the number of replicates used in each experiment, providing this information either in the figure legends, the Methods section, or preferably in both places.
- **Results/Figures:** Upon reviewing the results section, it appears that the author is performing multiple comparisons without statistically adjusting for them. To ensure appropriate statistical analysis, I recommend using an appropriate method such as ANOVA (or Bonferroni) instead of t-tests for determining significant differences among multiple groups. ANOVA is specifically designed for comparing means across multiple groups and would provide a more suitable approach for the statistical analysis conducted in the study. By using ANOVA or an equivalent method, the conclusions regarding the significance of differences would align with the appropriate analytical technique.
- **Discussion/Figure 6F:** I have noticed an inconsistency in the ratio of initial cells for the resveratrol cultivation experiment, particularly with strain AW_Res2, where both a 1:6 and a 20:1 ratio is shown as beneficial. To ensure clarity and address this observation, I kindly request the authors to provide further elaboration or clarification regarding this inconsistency.
- **Supplement information:** Upon considering the nature of the data and the presence of multiple comparisons in the "Model simulation and sensitivity analysis", it appears that using t-tests may not be the most appropriate statistical test. Given the potential for multiple testing, it is recommended to employ a suitable multiple comparison tests and apply a correction to account for this issue. Methods such as Bonferroni correction or false discovery rate (FDR) adjustment can be considered to address the increased risk of false positives. By incorporating these adjustments, the statistical analysis would better account for multiple comparisons and improve the reliability of the results. I suggest that the authors revise their analysis accordingly to ensure the appropriate statistical approach is applied.

Minor comments:

- **Line 49-50:** To maintain consistency in terminology and better align with established ecological concepts, I suggest that the authors consider using the term "mutualism" instead of "cooperation" and "parasitism" instead of "predation". This change would enhance clarity and ensure conformity with existing ecological literature.
- **Line 169:** For consistency and coherence, I recommend swapping the positions of uracil and methionine to align with the order presented in lines 168 and 171. This adjustment will enhance the clarity and organization of the information provided.
- **Line 196-197:** To provide a more comprehensive understanding, I kindly request the authors to elaborate on the specific criteria used to determine the strength of strain abilities in facilitating growth (i.e., strong/medium/weak). Additionally, I would like to

discuss the categorization of Lysine, as I believe it should be classified as medium based on the available information. Further clarification on these points would enhance the clarity and accuracy of the findings.

- Line 197: There appears to be a typo in the manuscript, where "tweak" is used instead of "weak." Please correct this error.
- Line 199-201: To ensure the robustness of the findings, it is essential to determine whether the observed differences in strain abilities were statistically significant. Therefore, I request the authors to provide information regarding the statistical analysis performed to confirm the overexpression of target genes. Including appropriate statistical tests and the corresponding p-values would strengthen the validity and reliability of the results presented in the study.
- Line 216-216: It appears that there are differences in the final optical density (OD) among some monoculture samples. To clarify the growth patterns, I kindly request the authors to provide information regarding the initial OD (t₀) and the OD at the final time point (t₄₈). If the OD at t₀ was not the same as the OD at t₄₈, please revise the statement from "no growth" to "limited growth" to accurately reflect the observed patterns in the monoculture samples. This clarification would enhance the accuracy and completeness of the reported results.
- Line 344-345: Was the difference between C_Res1 and Mctrl significant?
- Line 448: Please specify the ssDNA concentration used.
- Line 597: One is written in bold. Please update the text.
- Figure 3B: To maintain consistency in nomenclature, I suggest the authors follow a similar naming convention as used in Figure 3E. Ensuring uniformity in nomenclature allows for easier understanding and interpretation of the results by readers.
- Figure 4G: To improve clarity and consistency in the figures, I recommend moving the y-axis and x-axis title outside of the coloured triangle, similar to the format used in Figure 4E and Figure 4F. This adjustment will contribute to a more visually appealing and informative representation of the data in the figure.
- Figure S8 legend: Considering that the growth patterns are not identical, it is evident that the experimental factors (FPs) have some effect. However, to justify the claims it is crucial to emphasize that these effects did not result in any statistically significant differences. To provide a more comprehensive understanding, I recommend conducting statistical tests, such as ANOVA followed by a Tukey post hoc analysis, to explicitly assess and compare the effects of the FPs on growth. Employing these tests will allow for a thorough evaluation of any potential significant differences and provide valuable insights into the impact of the experimental factors.
- Figure S12 legend: To enhance clarity and improve the figure presentation, further description is necessary for the figure. Specifically, it would be helpful to provide an explanation of what the dots represent in relation to the bar plots. Currently, the dots appear to be overlapping with the bar plots, making it difficult to discern their meaning. Adding a stroke around the circle would solve this. Additionally, since there is ample space on the plot, I suggest improving the visual separation between data points and enhancing the breaks to make them more visually appealing and easier to interpret. Furthermore, it would be valuable to provide an explanation for why arginine production is higher with the native promoter and whether any of these strains exhibit a growth defect. Including information about the final optical density (OD) of these cultures would also provide important context for the observed results. Providing these details will contribute to a more comprehensive understanding of the findings presented in the figure.
- General: I have noticed a variation in the legend style used throughout the manuscript (supplement information). Specifically, the legends are presented in different formats, such as "(A)", "A.", and "A." in bold. To ensure consistency in the legend style, I recommend using a standardized format across all figures according to Nature's guidelines.
- Figure S13: Please provide a brief explanation of the technical issue that caused the drop in optical density (OD) in Figure S13A. Additionally, clarify whether this issue may have affected other quantifications in the study.

Reviewer #2 (Remarks to the Author):

This work by Peng et al. aimed to engineer a library of platform strains to create

Saccharomyces cerevisiae co-culture systems in which population dynamics are governed through metabolite exchange, which was identified as a key influential factor through global sensitivity analysis. The development of robust methodologies for co-culture bioproduction systems is highly relevant to metabolic engineering, and this manuscript contributes interesting insights to this research area. This toolkit might be useful for bioproduction processes where mono-cultures fail to deliver the desired performance due to the burden of maintaining cell growth and artificial metabolic transformation. The authors should address the following comments to improve this manuscript.

- Figure 2: What is the statistical significance between these groups (high/medium/low)? What are the criteria for classifying them as high/medium/low?
- Figure 2, 3, 4, 5: Please present the raw flow cytometry data used for the analysis of the percentage of subpopulations in all yeast co-cultures (mentioned in the Methods section, lines 524-526). Please also provide a detailed example of how the subpopulations were quantified from raw data, as I believe this analysis needs rigorous criteria for consistent interpretation across all the samples. I consider this data critical for evaluating the performance of co-cultures in combination with final OD values.
- Lines 208-209: One of the selling points of this work is that the authors made a toolkit for a yeast synthetic consortium comprising many types of auxotrophic markers and overproducers. However, in the experiments at lines 208-209, 7 out of 8 were paired with a lys strain without explanation. Please add some explanation of the reason why the authors chose lys for the assay.
- Line 230: The authors claimed that "This could suggest a limitation in the secretion-uptake-needs of his (H)." Please add data to show the concentration of his in the media to support the authors' claim.
- Lines 269 and 277: The authors mentioned that they chose the pair for the assay because the pairs showed different growth rates at a 1:1 ratio. However, at line 277, the results showed that ade-phe and ade-val showed almost equal proportion at a 1:1 ratio. Lines 269 and 277 are inconsistent. Please clarify.
- Figure 3F: Why is the growth rate of AKM_VIII much lower than AKM_IV? Could you add some explanation to this?
- Figure 4G, S20D: Based on the definition of BOD and ROD, the sum of BOD and ROD should be equal to OD and should not be higher than OD. However, in these figures, many ROD values are higher than the corresponding OD values. The authors need to further elaborate if the data is correct. If not, please correct the data analysis.
- Figure 5G: The data showed that the synthetic consortium's growth dynamics depend on both the initial OD and OD ratio of strains, which may suggest the system is highly sensitive to perturbations in the cell population, indicating potential instability and low reproducibility in the synthetic consortium's growth dynamics. However, it was not clear in the manuscript how many technical replicates were performed in independent experiments. To prove the reproducibility of the growth dynamics, which is important for a toolkit, please add data to show the assay's reproducibility.
- Figure 6: Since the co-culture systems were mostly characterized using OD700nm, applying this same method in the proof-of-concept experiment will demonstrate the platform's consistency and allow troubleshooting where necessary.
- Figure 6E, F: What is the statistical significance between these groups, and the samples in each group?
- Please label the wavelength of OD measurements (600 or 700) where relevant throughout the manuscript to improve clarity.
- Figure 6: It is confusing to use ade1-6 and lys1-6 as names of strains because they look like gene allele names. Please consider changing them to a different nomenclature, such as ade#1, to avoid confusion.

- Figure S8: Please include the wavelength of OD measurements (600 or 700) in all graphs in this manuscript to avoid confusion.

Reviewer #3 (Remarks to the Author):

The manuscript details the generation of a library auxotrophic and metabolite overproducing *S cerevisiae* strains. The work includes modelling to determine the how receptive this approach to creating synthetic microbial communities. There is a large amount of characterisation work performed and the demonstration of the systems use for division-of-labour.

The topic is important and this work makes a valuable contribution to the field. The experiments performed are good and well described in the manuscript.

Major comments:

The model is based on the model from Liao et al. 2020. In this manuscript, the authors have simplified the model in places but I believe these simplifications have made it slightly harder to read, removed some aspects that may be important, and reduced the reusability of the model. I understand the desire to "simplify" the model and reduce parameters but I believe this has been counter-productive.

1. The new model only includes metabolite uptake and use by non-producing strains. This is surely biologically inaccurate and reduces model generality. Why was this choice made?
2. The model is limiting strains to only be able to produce one metabolite for use by other strains. This is restrictive for future use where one might want to express multiple metabolites. Further, given that wild-type yeast can be quite leaky of their amino-acids into the environment, it doesn't seem biologically accurate either.
3. $J_{upt,j}$ in eq4 has a + rather than x in the numerator.
4. What is the reason for the removal of toxicity terms from eq6? From the data, it looks like this may be happening in some cases (see comment below regarding his)
5. "we parameterise the model as described in the methods" – this is not described in the methods. It is very difficult to follow how the initial model exploration was performed, what parameters were used etc.

Minor comments:

There is some mention of previous work using external control methods for microbial communities, but there is no mention of previous work on the kinds of self-regulating communities generated here. In particular:

10.1038/s41467-020-20756-2 for work on community design

10.1038/s41467-021-22240-x for experimental work, performing many of the same sort of characterisation methods described here.

Line 62 "Progress on establishing cross-feeding *E. coli* communities has been made" – references

Line 107 "it is" -> its

Line 119-120 "two strains, denoted $i = 1$ (producing metabolite 2) and $i = 2$ (producing metabolite 1" – Fig 1A shows J1 producing x1 not x2

Line 152 "traceable" -> tractable

Fig 4G – comment on the drop in the red population during stationary phase

Fig S16E – initial densities were 0.4 but max densities were 0.2. Please comment.

I believe there are multiple measures of OD being reported; OD600 and OD700 from a spectrophotometer and from a plate reader. This is currently making it difficult to compare certain data and follow changes e.g. Fig S16E, initial densities were 0.4 but max densities were 0.2. This should be easy to rectify by calibrating the machines so data can be reported on the same scales. See 10.1101/803239, 10.1021/acssynbio.0c00296.

Line 491 Greiner 655090 are cell-culture treated plates. Does this have any effect on yeast growth, protein/metabolite binding and availability etc.?

Fig 1 B – it is not clear if this modelling is performed for monocultures of co-cultures.

Fig 1B right – is this output at a particular time?

Fig 2D – I'm not sure I understand the categorisations of high, medium and low

Fig 5C – Adding his reduces the his auxotroph proportion? Please comment.

Author Rebuttal to Initial comments

We have addressed all comments and made several significant revisions, which we believe has elevated the manuscript into a more robust contribution to the field of synthetic microbial communities.

Key Updates:

- Conducted ANOVA along with Bonferroni's or Tukey's multiple comparison tests to enhance statistical rigor throughout the manuscript.
- Refined our mathematical model, including a detailed statistical analysis and explanation.
- Expanded our modelling approach to include exchange, toxicity and re-uptake of metabolites.
- Developed a more precise classification system for co-culture targets, supported by updated statistical analyses.
- Unified optical density measurements throughout the manuscript for better readability.
- Carried out an additional, independent set of tests on the three-member co-culture AKW_VI to confirm its reproducibility.
- Included new metabolite secretion and uptake data from LC-MS.
- Updated all main and supplementary figures, providing corresponding source data.
- Introduced a graphical abstract to summarise the study.
- Elaborated on our engineering strategy for the yeast co-culture toolkit in the main text.

Our revised manuscript now provides a clearer presentation of our results and conclusions. The updated manuscript serves as a comprehensive guide, detailing how various factors—such as metabolic exchange rates, population inoculation ratios, and population density—can be manipulated to influence community growth trajectories.

We are confident that our work will stimulate both applications and academic investigation of syntrophy in biomanufacturing settings. We trust you will find these revisions significantly enhance the quality of our manuscript.

Reviewer Expertise:

Referee #1: Synthetic biology, synthetic microbial communities

Referee #2: Synthetic communities, yeast

Referee #3: Modelling, synthetic biology

Reviewer Comments:

Reviewer #1 (Remarks to the Author):

The authors have developed a toolkit for creating co-cultures of *Saccharomyces cerevisiae*, an important industrial organism. They engineered yeast strains that can exchange essential metabolites, allowing the formation of multi-strain consortia. Through modelling and experiments, they investigate the factors influencing population dynamics in these co-culture systems. The toolkit is demonstrated to enhance and tune resveratrol production. Overall, this work provides a valuable resource for synthetic ecology and biomanufacturing applications.

We would like to thank reviewer 1 for their comments.

Major comments:

Whole manuscript: To ensure consistency and clarity throughout the manuscript, I suggest that the authors specify which wavelength of optical density (OD) measurement was used whenever referring to OD values. Since two different ODs are presented in the manuscript (600nm and 700nm), including the wavelength information alongside OD measurements would help avoid confusion and provide a clear understanding of the reported results. Furthermore, to adhere to the standard convention for writing optical density (OD), I recommend revising the manuscript to include the subscript denoting the wavelength whenever OD is mentioned.

We appreciate the comment and we have now always included the wavelength value as suggested. We also added wavelength-specific subscripts to all mentioned OD measurement in the manuscript.

In addition, and following the suggestion of another reviewer, in the revised manuscript we used OD_{700nm} values across all our results, enabling better comparability. For that, two standard curves were added to convert both OD_{600nm} values from a spectrophotometer and microplate reader into the OD_{700nm} equivalent of a Tecan microplate reader (Tab. S9).

For clarity and traceability, we have included in each raw data file the equipment used and the wavelength measured as well as the standard curve used for their conversion.

The only remaining instances of OD_{600nm} values using cuvette by spectrophotometer are in the Methods section, which may be convenient for other researchers to replicate our experiment.

Method section: To ensure clarity and consistency, I recommend that the authors specify the statistical methods used in the Methods section, including the specific tests employed (e.g., t-tests, ANOVA) and the corresponding software used for the statistical analysis. Additionally, it would be beneficial to include the number of replicates used in each experiment, providing this information either in the figure legends, the Methods section, or preferably in both places.

We appreciate this point, and in the revised manuscript, we have added details about our statistical methods, including the specific test names and corresponding software used. We have also included the number of technical replicates within both the figure legends and Methods section.

New Methods:

Statistical analysis and reproducibility

*Unless explicitly indicated, all data were subjected to analysis using Prism 9.5.0 (GraphPad) software. The error bars presented in the figures correspond to the standard deviation, as specified in the figure legend. Statistical analyses were conducted using either one-way or two-way ANOVA, followed by Tukey's or Bonferroni's multiple comparison test. Statistical significance, denoted as * when $p < 0.05$, was determined through these analyses: ns, not significant; *, $p < 0.05$; **, $p < 0.01$; ***, $p < 0.001$; ****, $p < 0.0001$. The figure legend provides information regarding the exact number of experimental replicates, all of which have been incorporated into the manuscript.*

Results/Figures: Upon reviewing the results section, it appears that the author is performing multiple comparisons without statistically adjusting for them. To ensure appropriate statistical analysis, I recommend using an appropriate method such as ANOVA (or Bonferroni) instead of t-tests for determining significant differences among multiple groups. ANOVA is specifically designed for comparing means across multiple groups and would provide a more suitable approach for the statistical analysis conducted in the study. By using ANOVA or an equivalent method, the conclusions regarding the significance of differences would align with the appropriate analytical technique.

We agree with the reviewer and have improved our statistical methods in the revised manuscript. We now perform ANOVA analysis, followed by Tukey's or Bonferroni's multiple comparison tests, to identify significant differences among distinct samples or groups (using the Prism 9.5.0 (GraphPad) software). We have updated the Results section, figures, and figure legends with this new statistical analysis.

→ See New Results, main Figure 1, Figure 2, Figure 3, Figure 5, Figure 6, and supplementary Figure S1-12, S14, Figure S18-20, Figure S28-30 and Figure S36.

Discussion/Figure 6F: I have noticed an inconsistency in the ratio of initial cells for the resveratrol cultivation experiment, particularly with strain AW_Res2, where both a 1:6 and a 20:1 ratio is shown as beneficial. To ensure clarity and address this observation, I kindly request the authors to provide further elaboration or clarification regarding this inconsistency.

This is an interesting point we also noticed but did not discuss in our previous version of the manuscript. The co-culture AW_Res2, comprising strain 1 (*FjTAL: trp⁺ade Δ*) and strain 2 (*At4CL_VvVSTI: trp Δ ade⁺*), was able to produce some of the highest amounts of resveratrol at ratios 1:6 or 20:1. The dynamics in bioproduction in co-culture with division of labour and cross-feeding are complex, since they are influenced by numerous factors: 1) as we have seen in Fig. S26, changes in inoculation ratio can massively change both growth rates of the two strains and final OD; 2) changes in growth rate are known to be accompanied by metabolic changes that can directly affect precursors and energy availability, which subsequently affects the flux in the pathways of interest and therefore production; 3) changes in the exchanged metabolites (auxotrophies and overexpressed genes) are also greatly affecting dynamics as well, as demonstrated in the variability of final ODs in Fig. S14; and 4) the order of the pathway split in relation to the syntrophic relationship (e.g. AW_Res1 vs AW_Res2).

In this particular case the differences between AW_Res2 at 1:6 and 20:1 are most likely a combination of factors 1 and 2. In the case of 1:6 ratio the final OD is slightly higher than in 20:1, suggesting that at least one of the growth rates will be different, which will change metabolism. It is possible that the ratio 1:6 which grows slightly higher has slightly less optimal metabolic fluxes than 20:1 – the advantage of one could be a higher amount of producer cells and the other an optimal production per cell, which in the end can lead to similar production levels even if initial ratios are different.

We have observed similar behaviours in our previous work, where we used simple knock-out based syntrophy for bioproduction using different initial ratios (<https://www.nature.com/articles/s41589-023-01341-2>). In that work we produced malonic semialdehyde (MSA) and found that the cross-feeding pairs his2-met3 and met14-trp4 showed higher production at extreme ratios (either 20:1, 10:1 or 1:10) than at intermediate ratios (5:1 or 1:2) – figure below (light pink and light purple).

Figure 5 panel e from Aulakh et al Nat Chem Bio 2023

We have discussed the results of this particular co-culture and its inoculation ratio-independent phenotype in the Results section, where we comment on how such a combination of asymmetric growth rates, metabolic and cross-feeding constraints can enable robust product formation in batch culture.

→ New Results on lines 374-376:

“Interesting, AW_Res2 also achieved a similarly high resveratrol production at initial ratio 20:1, which suggests the importance of the combination of differences in growth rates, metabolic and cross-feeding constraints in bioproduction.”

Supplement information: Upon considering the nature of the data and the presence of multiple comparisons in the “Model simulation and sensitivity analysis”, it appears that using t-tests may not be the most appropriate statistical test. Given the potential for multiple testing, it is recommended to employ a suitable multiple comparison tests and apply a correction to account for this issue. Methods such as Bonferroni correction or false discovery rate (FDR) adjustment can be considered to address the increased risk of false positives. By incorporating these

adjustments, the statistical analysis would better account for multiple comparisons and improve

the reliability of the results. I suggest that the authors revise their analysis accordingly to ensure the appropriate statistical approach is applied.

We appreciate this point raised by the reviewer. During our model simulation and sensitivity analysis, the eFAST algorithm generates sensitivity indices for all parameters. However, this can introduce artifacts, for example, including parameters in the eFAST analysis which are not present in the model will receive a small, non-zero, sensitivity score. Therefore, Marino *et al.* proposed a method which takes advantage of this fact: by including such a parameter in the analysis (i.e., the dummy parameter which we call δ) we can determine a threshold sensitivity. We used a t-test to compare all other parameters in the model to this dummy parameter. We see the reviewer's concern that some of our models involve comparing the results of multiple parameters to this dummy parameter. Therefore, we have repeated our analysis using Bonferroni correction and a more conservative alpha value of 1%. We have amended the supplementary material as follows:

"eFAST settings. The sensitivity analysis was run using 100 re-samplings with 1,285 samples per search and 4 Fourier coefficients retained. We set a p-value threshold for significance of 1% but used Bonferroni correction to account for multiple testing. The significance threshold for each analysis is therefore $0.01/n_k$ where n_k is the number of parameters varied in that analysis."

→ See **Revised Figure 1 and 6** and **Supplementary Note 1-4**. All eFAST sensitivity figures have been updated to highlight where sensitivity and total sensitivity are statistically significant.

Minor comments:

Line 49-50: To maintain consistency in terminology and better align with established ecological concepts, I suggest that the authors consider using the term "mutualism" instead of "cooperation" and "parasitism" instead of "predation". This change would enhance clarity and ensure conformity with existing ecological literature.

We agree with the reviewer. In the revised manuscript we have changed the terms "mutualism" and "parasitism" to "cooperation" and "predation," respectively.

Line 169: For consistency and coherence, I recommend swapping the positions of uracil and methionine to align with the order presented in lines 168 and 171. This adjustment will enhance the clarity and organization of the information provided.

The order of uracil and methionine has been swapped on line 169 (previous line 168) in the revised manuscript as suggested.

Line 196-197: To provide a more comprehensive understanding, I kindly request the authors to elaborate on the specific criteria used to determine the strength of strain abilities in facilitating growth (i.e., strong/medium/weak). Additionally, I would like to discuss the categorization of Lysine, as I believe it should be classified as medium based on the available information. Further clarification on these points would enhance the clarity and accuracy of the findings.

We have revised the manuscript to better delineate what constitutes strong/medium/weak capacity to promote co-culture growth. Also, as mentioned above, we have redone our statistical analysis with more rigorous ANOVA testing, followed by Bonferroni's multiple comparisons tests. We have now selected arbitrary thresholds for the definition of strong ($OD_{700nm} \geq 0.5$), medium ($0.3 \leq OD_{700nm} < 0.5$) and weak ($OD_{700nm} < 0.3$) which has consequently reclassified the co-cultures in Figure 2D. Notably, this reclassification has moved lysine into the "medium" category, as opposed to its previous classification of "high" or "strong", and in accordance with reviewer's opinion.

*Based on the growth (OD_{700nm}) the co-cultures, we classified each target metabolite by their ability to facilitate growth in cross-feeding co-cultures: strong ($OD_{700nm} \geq 0.5$): *ade, trp, met, his*; medium ($0.3 \leq OD_{700nm} < 0.5$): *lys, phe&tyr, val&ile, cys, leu, ura*; and weak ($OD_{700nm} < 0.3$): *thr, tyr, arg, ser*.*

→ See **Main Text**, **Revised Figure 2**, and **Supplementary Note 5**.

Line 197: There appears to be a typo in the manuscript, where "tweak" is used instead of "weak." Please correct this error.

We have amended this typo in the revised manuscript.

Line 199-201: To ensure the robustness of the findings, it is essential to determine whether the observed differences in strain abilities were statistically significant. Therefore, I request the authors to provide information regarding the statistical analysis performed to confirm the overexpression of target genes. Including appropriate statistical tests and the corresponding p-values would strengthen the validity and reliability of the results presented in the study.

We appreciate the reviewer's comment. As mentioned above, we have performed ANOVA, followed by Bonferroni's multiple comparisons test on the metabolites production data from LC-MS, this is reflected in the revised Fig. S18 (previous Fig. S12). This figure has also undertaken a substantial revision. Revisions entail the division of each exchanged metabolite into an individual panel, accompanied by cell growth and strain identification. Additionally, the raw data (including p-values) of ANOVA tests, followed by Bonferroni's multiple comparisons tests can be found in the Source data (Fig. S18).

→ See **Supplementary Figure S18** (previous **Figure S12**) and **Source Data**.

Line 216-216: It appears that there are differences in the final optical density (OD) among some monoculture samples. To clarify the growth patterns, I kindly request the authors to provide information regarding the initial OD (t_0) and the OD at the final time point (t_{48}). If the OD at t_0 was not the same as the OD at t_{48} , please revise the statement from "no growth" to "limited growth" to accurately reflect the observed patterns in the monoculture samples. This clarification would enhance the accuracy and completeness of the reported results.

We thank the reviewer for pointing out this issue. Indeed, some monocultures have final time point OD_{700nm} values that are different than the initial OD_{700nm} values. We have thus revised the statement from "no growth" to "limited growth" in the main text. To better reflect co-culture growth potential, we have replaced the OD data (48 hours) in Fig. 3C and Fig. 3F with the maximum OD_{700nm} within 72 hours of cell culture. We have also added the initial OD values

(OD_{700nm} 0 hours) and the maximum OD values within 72 hours in the new supplementary

Figure S20.

Line 344-345: Was the different between C_Res1 and Mctrl significant?

Yes. Based on the statistical analysis results of ANOVA (**Fig. 6**), the difference of OD_{700nm} at 48 hours between C_Res1 and Mctrl was significant ($p < 0.0001$). We revised “higher” to “significantly higher” in the revised manuscript.

Line 448: Please specify the ssDNA concentration used.

The concentration of ssDNA from Invitrogen (Catalog No.15632011) was 10 mg/mL, and we have added this information into the revised manuscript.

Line 597: One is written in bold. Please update the text.

We have corrected this typo.

Figure 3B: To maintain consistency in nomenclature, I suggest the authors follow a similar naming convention as used in Figure 3E. Ensuring uniformity in nomenclature allows for easier understanding and interpretation of the results by readers.

We have revised the nomenclature in Figure 3B to be of similar naming convention to that used in Figure 3E.

Figure 4G: To improve clarity and consistency in the figures, I recommend moving the y-axis and x-axis title outside of the coloured triangle, similar to the format used in Figure 4E and Figure 4F. This adjustment will contribute to a more visually appealing and informative representation of the data in the figure.

We have updated the position of the x- and y-axis labels in the revised Figure 4G as suggested.

Figure S8 legend: Considering that the growth patterns are not identical, it is evident that the experimental factors (FPs) have some effect. However, to justify the claims it is crucial to emphasize that these effects did not result in any statistically significant differences. To provide a more comprehensive understanding, I recommend conducting statistical tests, such as ANOVA followed by a Tukey post hoc analysis, to explicitly assess and compare the effects of the FPs on growth. Employing these tests will allow for a thorough evaluation of any potential significant differences and provide valuable insights into the impact of the experimental factors.

We have now added statistical tests (two-way ANOVA followed by Bonferroni’s multiple comparisons test) to the data in Fig. S11B-C (previous Fig. S8B-C where “ns” = not significant and “*” = $P < 0.05$). We calculated the significances of the observed differences in OD_{700nm}, of the expression of the 3 FPs (GFP, BFP and RFP) in three media YPD, SD, SM, at three different time points: 10 h (end of exponential phase in YPD and SD), 15.5 h (end of exponential phase in SM) and 48 h (end of the experiment) (revised Fig. S11B,C below and Source data Fig. S11). We found no significant differences between groups at 10 h and 15.5 h. We found that only BFP vs. Ctrl in YPD and RFP vs. Ctrl in SM, showed statistically

significant differences at 48 h. However, it is important to note that the influence of the

differences appear to be quite limited, since the overall variations in OD700 are small. In addition, we also calculated the growth rates of the strains expressing the different FPs in the different media and used the same statistical test described above. We found that there were no significant differences in the growth rates (revised Fig. S11D below). These observations suggest that the expression of GFP, BFP, and RFP had minimal discernible impact on the yeast growth trajectories within the investigated time frames and under the specified culture conditions.

→ See Revised Supplementary Figure S11 (previous Fig. S8)

Figure S12 legend: To enhance clarity and improve the figure presentation, further description is necessary for the figure. Specifically, it would be helpful to provide an explanation of what the dots represent in relation to the bar plots. Currently, the dots appear to be overlapping with the bar plots, making it difficult to discern their meaning. Adding a stroke around the circle

would solve this.

Additionally, since there is ample space on the plot, I suggest improving the visual separation between data points and enhancing the breaks to make them more visually appealing and easier to interpret.

Furthermore, it would be valuable to provide an explanation for why arginine production is higher with the native promoter and whether any of these strains exhibit a growth defect. Including information about the final optical density (OD) of these cultures would also provide important context for the observed results. Providing these details will contribute to a more comprehensive understanding of the findings presented in the figure.

We agree with the reviewer, and we have undertaken a substantial revision for **Figure S18** (previous **Figure S12**). Revisions entail the division of each exchanged metabolite into an individual panel, accompanied by the inclusion of optical density (OD), strain identification, and additional statistical analysis. In the revised **Figure S18**, grey bar represents OD_{700nm} at 23 hours, blue bar represents the concentration of exchanged metabolites (mg/L), and the error bars correspond to the standard deviation. Each data point was represented by a circle/point in the graph. The revised **Figure S18** shows how the expression of the 15 exchanged metabolites impacted both cellular growth and metabolite production.

Regarding the metabolite arginine (Fig. S18B), we did observe a reduction of arginine production (with no effect on cell growth) when a stable form of *MPRI* (MRP1 G85E) was overexpressed, contrary to our initial expectations. *MPRI* had been found to acetylate the proline metabolism intermediate Δ^1 -pyrroline-5-carboxylate (P5C)/glutamate- γ -semialdehyde (GSA) into N-acetyl-GSA for arginine synthesis in the mitochondria (Nishimura et al. 2010). We selected this target because the expression of the *MPRI* G85E variant was reported to increase l-arginine synthesis in yeast (Nasuno et al. 2016). This effect was not found in our case, which could be due to the fact that the strain used by Nasuno et al was metabolically and genetically distinct from the strain we used (BY4741) or because Nasuno et al used a high-copy plasmid expression system while we used genome integration (leading to different *MPRI* levels and therefore differential metabolic effects). The effect we observed could be caused by an excess accumulation of N-acetyl-GSA, which is known to inhibit the downstream arginine synthesis pathway (Nishimura et al. 2010; Takagi et al. 2019).

→ See Revised Supplementary Figure S18.

References

- Akira Nishimura and others, An antioxidative mechanism mediated by the yeast N-acetyltransferase Mpr1: oxidative stress-induced arginine synthesis and its physiological role, *FEMS Yeast Research*, Volume 10, Issue 6, September 2010, Pages 687–698, <https://doi.org/10.1111/j.1567-1364.2010.00650.x>
- Ryo Nasuno and others, Structure-based molecular design for thermostabilization of N-acetyltransferase Mpr1 involved in a novel pathway of L-arginine synthesis in yeast, *The Journal of Biochemistry*, Volume 159, Issue 2, February 2016, Pages 271–277, <https://doi.org/10.1093/jb/mvv101>
- Hiroshi Takagi, Metabolic regulatory mechanisms and physiological roles of functional amino acids and their applications in yeast, *Bioscience, Biotechnology, and Biochemistry*, Volume 83, Issue 8, 3 August 2019, Pages 1449–1462, <https://doi.org/10.1080/09168451.2019.1576500>

General: I have noticed a variation in the legend style used throughout the manuscript (supplement information). Specifically, the legends are presented in different formats, such as "(A)", "A.", and "A." in bold. To ensure consistency in the legend style, I recommend using a standardized format across all figures according to Nature's guidelines.

We have homogenized the legend style in the main and supplementary figures.

Figure S13: Please provide a brief explanation of the technical issue that caused the drop in optical density (OD) in Figure S13A. Additionally, clarify whether this issue may have affected other quantifications in the study.

As suggested by the reviewer, the drop in OD in Fig. S15A is a technical issue that is rarely observed in microplate readers, especially Tecan, and which is more likely seen in long experiments, with low working volumes and when the cells grow fast and reach high cell density – which promotes sedimentation especially when condensation/evaporation occurs (affecting the ability of the plate reader to accurately perform OD measurements). This is the case for Fig S15 panel A, which was a long experiment (72h) and the OD reached was the highest and fastest of all the other strains/panels. We have used the same strain in similar conditions in other experiments (e.g., Fig S11) and we have not observed that drop, indicating that it is indeed a isolated technical issue (see figure below).

Fortunately, this technical issue can be easily spotted by the drastic drop in OD and by sedimentation in the wells. In this case, despite the drop, it still shows the fast growth rate and high OD reached compared to other co-cultures in FigS15. We have now indicated in the figure caption: “the observed drop in OD is a technical issue linked to cell sedimentation observed at high cell densities in small volumes”. If preferred by the editorial team, we can substitute it by a replicate experiment where the issue did not happen.

We have also reviewed all the growth curves of the work and have not found such drastic OD decrease in other samples. While we believe that this effect does not affect the co-culture experiments since they grow slower and sedimentation (drastic OD drop) was not observed, in order to minimise potential minor effects in other quantifications, we have now reported maximum OD₇₀₀ throughout the manuscript, when possible and specially in long experiments (>48 h, Fig S14, S20, S28-30), instead of comparing OD₇₀₀ at certain time point.

Reviewer #2 (Remarks to the Author):

This work by Peng et al. aimed to engineer a library of platform strains to create *Saccharomyces cerevisiae* co-culture systems in which population dynamics are governed through metabolite exchange, which was identified as a key influential factor through global sensitivity analysis. The development of robust methodologies for co-culture bioproduction systems is highly relevant to metabolic engineering, and this manuscript contributes interesting insights to this research area. This toolkit might be useful for bioproduction processes where mono-cultures

fail to deliver the desired performance due to the burden of maintaining cell growth and

artificial metabolic transformation. The authors should address the following comments to improve this manuscript.

We thank reviewer 2 for their valuable comments and for considering the insights of our work of relevance. We have now addressed the comments as described below and improved the manuscript accordingly.

Figure 2: What is the statistical significance between these groups (high/medium/low)? What are the criteria for classifying them as high/medium/low?

We thank the reviewer for this comment. We have redone our statistical analysis with more rigorous ANOVA testing, followed by Bonferroni's multiple comparisons tests. We have now also indicated the arbitrary thresholds used for the definition of strong ($OD_{700nm} \geq 0.5$), medium ($0.3 \leq OD_{700nm} < 0.5$) and weak ($OD_{700nm} < 0.3$).

→ See **Main Text, Revised Figure 2, and Supplementary Note 5.**

*Based on the growth (OD_{700nm}) of the co-cultures, we classified each target metabolite by their ability to facilitate growth in cross-feeding co-cultures: strong ($OD_{700nm} \geq 0.5$): *ade, trp, met, his*; medium ($0.3 \leq OD_{700nm} < 0.5$): *lys, phe&tyr, val&ile, cys, leu, ura*; and weak ($OD_{700nm} < 0.3$): *thr, tyr, arg, ser*.*

Figure 2, 3, 4, 5: Please present the raw flow cytometry data used for the analysis of the percentage of subpopulations in all yeast co-cultures (mentioned in the Methods section, lines 524-526). Please also provide a detailed example of how the subpopulations were quantified from raw data, as I believe this analysis needs rigorous criteria for consistent interpretation across all the samples. I consider this data critical for evaluating the performance of co-cultures in combination with final OD values.

Thanks for the comment. We have now uploaded the raw flow cytometry (FC) data from Figure 4F and Figure 5C, 5F (Figures 2, 3 don't have FC data) on <https://github.com/hdpeng89/Raw-flow-cytometry-data-yeast-co-culture>, as described in the data availability. Following the comment of the reviewer, we have now added the Figure S37 to the manuscript to explain the method used to identify subpopulation using the FC with one co-culture as an example. We have also added one sentence in the Methods section of the revised manuscript to guide the reader to Fig. S37 where the details on the subpopulation analysis using FC are given: “*The detailed gate strategy of these flow cytometer data was shown as Fig. S37.*”

Fig. S37 Gating strategy for flow cytometry data.

Sample co-culture AKW_VI without exchanged metabolite supplementation (AKW_VI em_0, Figure 5 F) was selected as an example to demonstrate the data gating strategy.

A. *Yeast cells were gated for singlets using FSC-H vs FSC-A and to remove background noise. 29,187 (> 10,000) events were collected and analysed within the singlets gate*

for each measurement. **B.** Quadrant gating was applied in the double-fluorescence dimension (BL1-H::sfGFP vs YL2-H::mScarlet-I) to separate the population tagged with different fluorescent proteins including mscarlet-I (38.1%), sfGFP (18.2%), mTagBFP2 and non-fluorescent cells (42.3%), and multiple cells (1.43%) such as

doublet, triplet. **C.** Quadrant gating was further applied in double-fluorescence dimension (BL1-H::sfGFP vs VL 1-H::mTagBFP2) to separate the population tagged with mTagBFP2 (96% out of 42.3%) and non-fluorescent cells (4% out of 42.3%), then the percentage of mTagBFP2 tagged population was $96.0\% \times 42.3\% = 38.98\%$, and the percentage of non-fluorescent cells was $4.01\% \times 42.3\% = 1.69\%$.

Lines 208-209: One of the selling points of this work is that the authors made a toolkit for a yeast synthetic consortium comprising many types of auxotrophic markers and overproducers. However, in the experiments at lines 208-209, 7 out of 8 were paired with a lys strain without explanation. Please add some explanation of the reason why the authors chose lys for the assay.

We appreciate this comment by Reviewer 2. Due to the many different possible combinations for co-cultures we arbitrary selected lys to establish additional synthetic consortia. We were encouraged to select lys because of the good performance of the validated ade-lys coculture (Fig S12) and the improved secretion capacity of the lys+ modification (Fig S18). We have now removed the pair ade-trp from this section since this co-culture is already described in the previous section (section 2).

We have now added this rationale into the revised manuscript:

“We used auxotrophic/overproducer strains from our toolkit to create additional syntrophic co-cultures composed of two or three members. Having previously validated co-cultures with *ade*, we arbitrary decided to test co-cultures with *lys* (which performed well in the *ade-lys* co-culture) and we established *his-lys*, *leu-lys*, *phe-lys*, *trp-lys*^{v1}, *trp-lys*^{v2}, *val-lys*^{v1}, and *val-lys*^{v2} co-cultures (Described in **Fig. 3** and its caption), which displayed significantly higher cell growth than their monoculture controls (**Fig. 3A-C**).”

Line 230: The authors claimed that "This could suggest a limitation in the secretion-uptake-needs of *his* (*H*)." Please add data to show the concentration of *his* in the media to support the authors' claim.

We appreciate the reviewer's comment. We initially thought that the low levels of histidine found in the media of adenine auxotrophic strains with and without *HIS1* overexpression (former Fig. S12 new Fig. S18E) could support the observed results. To validate the generality of such result, we measured histidine levels with LC-MS in the media of lysine auxotrophic strains with and without *HIS1* overexpression and found higher amount (8.9-30 mg/L) (Fig S18E), suggesting that our previous hypothesis was not necessarily correct. We have therefore removed the sentence “*This could suggest a limitation in the secretion-uptake-needs of his (H)*” from the new version of the manuscript.

In addition we have now looked at the growth of the 3 members co-cultures and their populations over time (new Fig. S21A-E). We found that all the three strains in each of the *his*-related co-cultures AKH_III and HKW_V displayed growth, but it was very limited in all cases.

→ See new added supplementary Figure S21 below.

Fig. S21 Time courses of the cell growth of nine three-member co-cultures and their respective individual members.

Lines 269 and 277: The authors mentioned that they chose the pair for the assay because the pairs showed different growth rates at a 1:1 ratio. However, at line 277, the results showed that ade-phe and ade-val showed almost equal proportion at a 1:1 ratio. Lines 269 and 277 are inconsistent. Please clarify.

We thank the reviewer for pointing this out and we realize now that our previous description lacked clarity. These four co-culture pairs were actually selected as examples based on observed differences in their co-culture growth dynamics (Fig. S26, 1:1 ratio, middle column), rather than in the growth / ratio of each strain in the co-culture. In the case of ade-phe and ade-val co-cultures, despite exhibiting nearly equal proportions in a 1:1 ratio, they showed different growth pattern. The co-culture ade-val had a faster growth towards stationary phase (occurring at approximately 20 hours). In contrast, the ade-phe co-culture reached stationary phase around 32 hours, displaying a comparatively slower growth rate. The other two co-cultures were selected for having limited (ade-tyr) or not significant (ade-arg) growth at 1:1 inoculation ratio.

To accurately reflect this distinction, we have revised line 289, replacing “each displayed different growth” with “each co-culture pair displayed different growth dynamics”.

Figure 3F: Why is the growth rate of AKM_VIII much lower than AKM_IV? Could you add some explanation to this?

While both three-member co-cultures AKM_IV and AKM_VIII, are formed by strains that rely on the exchange of the metabolites adenine, lysine, and methionine, they differ significantly in terms of strain composition and communication patterns. In the co-culture AKM_IV, each strain is auxotrophic for one exchanged metabolite, necessitating complementation by another strain, which enables the establishment of what we have called ‘one-way communication’ among the three members. Conversely, the co-culture AKM_VIII consists of three strains, each with a dependency on two exchanged metabolites (double auxotrophs), and therefore relying on the other two members to supply the missing two metabolites, which results in what we have called ‘two-way communication’ among the three members.

The two-way communication system found in AKM_VIII requires a higher amount of exchanged metabolites (each sufficient to support the growth of 2 strains) and in addition, two members of the co-culture compete for the uptake the each exchanged metabolite, creating more complex conditions, which could explain the overall reduced community growth. The reduced growth on the two ways communication is also found when comparing AKW_VI with AKW_I but not with AKW_II, suggesting that the complexity of the dynamics is case specific and requires further investigation out of the scope of this work. Overall, the distinct strain compositions and communication mechanisms in these co-cultures contribute to their divergent growth dynamics and highlight the importance of considering such factors in designing and interpreting multi-member co-culture systems.

Figure 4G, S20D: Based on the definition of BOD and ROD, the sum of BOD and ROD should be equal to OD and should not be higher than OD. However, in these figures, many ROD values are higher than the corresponding OD values. The authors need to further elaborate if the data is correct. If not, please correct the data analysis.

We thank the reviewer for pointing out this issue. After conducting a careful review of the analysed data, we noticed that we missed the normalisation step in the BOD and ROD values presented in the original Fig. 4G, as well as the ROD value in Fig. S26D (previous Fig. S20D). As a result, we have rectified these errors and incorporated the corrected analysed data in the revised versions of Fig. 4G and Fig. S26D. The new figures are shown below. Furthermore, we have double-checked the analysis of OD/BOD/ROD/GOD data in all figures of the manuscript.

→ See **Revised Figure 4G**.

→ See Revised Figure S26D (previous Fig.S20D).

Figure 5G: The data showed that the synthetic consortium's growth dynamics depend on both the initial OD and OD ratio of strains, which may suggest the system is highly sensitive to perturbations in the cell population, indicating potential instability and low reproducibility in the synthetic consortium's growth dynamics. However, it was not clear in the manuscript how many technical replicates were performed in independent experiments. To prove the reproducibility of the growth dynamics, which is important for a toolkit, please add data to show the assay's reproducibility.

In our study, experiments were conducted in triplicates, and we have now included this information in the legend of the revised Figure 5 (and other figures legends across the

manuscript). The original data of each replicate can be found in Source data Fig. 5.

Furthermore, encouraged by the comment of the reviewer and to validate the reproducibility of the results, we have performed an additional independent set of repeated cell cultures for the co-culture shown in Fig 5G, AKW_VI, at four initial OD₇₀₀ values 0.067, 0.078, 0.102 and 0.148. This new set comprised three replicates, and the results are presented in the figure below (Source data Fig. 5). The consistency observed in the growth dynamics of co-culture AKW_VI across these replicates attests to the reproducibility of our experimental outcomes.

Original submission for Fig. 5G

Repeated data for Fig. 5G

Figure 6: Since the co-culture systems were mostly characterized using OD_{700nm}, applying this same method in the proof-of-concept experiment will demonstrate the platform's consistency and allow troubleshooting where necessary.

We agree with the reviewer and have now generated a standard curve (Tab. S9; below) by correlating the OD_{700nm} and OD_{600nm} measurements using the same Tecan plate reader. In the revised Figure 6, we have subsequently converted the OD_{600nm} values to OD_{700nm} units, aligning them with the standardized measurement scale. This adjustment ensures the consistency of the data representation in the figure. The new figure legend now refers to this OD conversion for transparency “OD_{700nm} values (calculated using Tab S9 for consistency)”.

Figure 6E, F: What is the statistical significance between these groups, and the samples in each group?

Thanks for this comment. We have now used Prism 9.5.0 (GraphPad) software to conduct statistical analyses: two-way ANOVA, followed by Tukey's multiple comparison test in the new Figure 6E and F. The analysed data is available in the Supplementary Source data.

About Figure 6E: Most co-culture group comparisons on the OD_{700nm} values at 48 hours are significant ($p < 0.0001$) except five group comparisons: C_Res1 vs. AK_Res1, AK_Res1 vs. AW_Res2, AK_Res1 vs. AW_Res2, AK_Res2 vs. AW_Res1, and WK_Res1 vs. WK_Res2. While OD_{700nm} values in two co-culture groups of both C_Res1 and WK_Res1 are not significantly different between different initial ratios, most OD_{700nm} values in other co-culture groups were significantly different. This indicated that both the initial ratio and the co-culture pair affect the growth of some co-culture groups.

About Figure 6F: While one subset of the nine co-culture group comparisons (C_Res1 vs. AK_Res1, C_Res1 vs. AW_Res1) showed no significant differences, the majority of groups showed significantly different resveratrol production at 48 hours ($p < 0.05$). Furthermore, with the exception of the co-culture WK_Res1, all co-culture groups had instances where distinct initial inoculation ratios led to significantly different resveratrol production at 48 hours. This suggests that both the initial ratio and the co-culture pair can generally affect resveratrol production in co-cultures.

Please label the wavelength of OD measurements (600 or 700) where relevant throughout the manuscript to improve clarity.

We have added wavelength information to all OD values in the main text, figures, and supplementary materials.

Figure 6: It is confusing to use ade1-6 and lys1-6 as names of strains because they look like gene allele names. Please consider changing them to a different nomenclature, such as ade#1, to avoid confusion.

We appreciate this suggestion by the reviewer and have now used the proposed nomenclature (incorporating the "#" symbol) to the strain names not only in Fig. 6 but also throughout the revised manuscript.

Figure S8: Please include the wavelength of OD measurements (600 or 700) in all graphs in this manuscript to avoid confusion.

We have added wavelength information to OD values in Figure S11 (previous Figure S8). As mentioned above, we have also added wavelength information to all OD values found in the revised main text, figures, and supplementary material.

Reviewer #3 (Remarks to the Author):

The manuscript details the generation of a library auxotrophic and metabolite overproducing *S cerevisiae* strains. The work includes modelling to determine the how receptive this approach to creating synthetic microbial communities. There is a large amount of characterisation work performed and the demonstration of the systems use for division-of-labour. The topic is important and this work makes a valuable contribution to the field. The experiments performed are good and well described in the manuscript.

We would like to thank Reviewer 3 for highlighting the importance of the work, the large amount of work, and the quality of the experiments and their description in the text.

Major comments:

The model is based on the model from Liao et al. 2020. In this manuscript, the authors have simplified the model in places but I believe these simplifications have made it slightly harder to read, removed some aspects that may be important, and reduced the reusability of the model. I understand the desire to “simplify” the model and reduce parameters but I believe this has been counter-productive.

Thanks for the comment. Liao *et al.* produced a comprehensive and predictive model of *E. coli* co-culture dynamics which required extensive data for accurate calibration. In this work, we chose to focus on identifying the key “engineering dials” (i.e. experimental engineering approaches) which contribute to consortia dynamics via a global sensitivity analysis approach. Therefore, we decided to simplify the original model (for example, removing the equations which captured toxicity or chemostat growth) to focus on the core principles of the consortia interactions in batch culture. These simplifications enabled us to focus on determining the most important aspects of the consortia to engineer. We agree with the reviewer that this approach means that we are not capable of *predicting* the performance of *specific* consortia which is a potential limitation of our “model first” approach. However, this approach is relatively “data free” enabling us to identify engineering interventions before undertaking experimental work. Guided by the appropriate comment of the reviewer we have now improved the communication of the model and its notation; we have made a number of alternations to Supplementary Note 1 and the main text. For example, we have clarified that x_i represents the concentration of exchange metabolite i which is produced by strain y_i . We have clarified the meanings of the J notation as representing rates. We have expanded our overall description of our modelling

methods giving more information of parameter selection which we hope conveys our approach more clearly. To ensure that the details of each model are clear we have updated Supplementary

Notes 2, 3 and 4 to explicitly give the ordinary differential equation models for two-member and three member co-cultures.

The new model only includes metabolite uptake and use by non-producing strains. This is surely biologically inaccurate and reduces model generality. Why was this choice made?

We appreciate the comment. We initially assumed that if production strains were engineered by overexpressing key biosynthetic genes in a constitutive manner (and avoiding feedback inhibition in most cases) then any excretion would be that above the requirements for growth and so it would be unlikely that our overproduction strains would actively take up the metabolite they produced. We find this holds for most of our strains (Fig. S18-19) but agree with the reviewer that uptake of a metabolite by its overproduction strain could have impacts on the dynamics of our co-culture systems. Therefore, we developed an updated model which includes uptake of the exchange metabolite by the overproduction strain and reported the results in the new **Part 4 of Supplementary Note 2** and **Part 4 of Supplementary Note 3**. We find that these new processes do not change the sensitivity analysis significantly and any changes are fully reported in their new Parts and new **Figure S4** and **S8**.

The model is limiting strains to only be able to produce one metabolite for use by other strains. This is restrictive for future use where one might want to express multiple metabolites. Further, given that wild-type yeast can be quite leaky of their amino-acids into the environment, it doesn't seem biologically accurate either.

We agree with the reviewer that WT yeast often 'over produce' amino acids creating a leak of amino acids into the environment; in our study we see this in our co-cultures containing methionine and tryptophan, where both ade-met IV, ade-trp IV co-cultures showed reasonable growth without excess expression of methionine or tryptophan (Figure S14). We discuss this in Supplementary Note 3 (specifically Part 2) and present these results in Figure S5. In this section, we updated the model to include a second "leak reaction" which creates a second bidirectional communication between these strains. We found that this model with a second metabolite exchange reaction broadly recapitulates our findings where this reaction is excluded. However, we do see that the final population of the consortia and glucose uptake rates show significant sensitivity to the strength of this second leak reaction.

$J_{\text{upt},j}$ in eq4 has a + rather than x in the numerator.

We thank the reviewer for identifying this typographical error which has now been corrected.

What is the reason for the removal of toxicity terms from eq6? From the data, it looks like this may be happening in some cases (see comment below regarding his)

Our initial modelling aims were to identify how different parameters contributed to the behaviour of co-culture systems. We chose to focus on core system interactions and therefore we neglected toxicity. However, as the reviewer notes toxicity can arise in these systems and our exclusion of this reduces the application of our observations to other systems. Therefore, we have now modified our model to account for toxicity and re-analysed the model using the global sensitivity analysis approach. We find that whilst the inclusion of toxicity does reduce the sensitivity, it does not significantly change the rank order of the parameters (i.e., our initial observations of process importance hold). We have updated Supplementary Note 2 as follows:

(1) We first describe the updated model. To account for toxicity, we modified Eq. 6 to new Eq. 14:

$$J_{i,t}^* = \min(J_{i,t}^*, J_{i,t}^* \cdot T_{i,t}^*)$$

where $T_{i,t}^*$ captures the impact of toxicity in a similar manner to the original Liao et al. study.

(2) We discuss these new results in Supplementary Note 2 Part 3 “Global sensitivity analysis of a two-strain co-culture with toxicity of the metabolites included.”

→ See **Revised Figure S3**

Additionally, regarding your last comment “Fig 5C – Adding his reduces the his auxotroph proportion?” We don’t think the his addition added toxicity to the yeast cells (Fig. S27D), the slight reduction of the his auxotroph proportion was because of the faster growth rate of ade auxotrophic strain (tagged with RFP) in the co-culture. Please refer to the responses to the last comment for detailed explanation.

“we parameterise the model as described in the methods” – this is not described in the methods. It is very difficult to follow how the initial model exploration was performed, what parameters were used etc.

We thank the reviewer for noting the model parameter has been omitted. Supplementary Note 1 has been now updated as follows:

“Parameter ranges for eFAST. The specific parameters varied in each sensitivity analysis are reported in the respective figures and full results are shown in the Supplementary Figures. In the eFAST analysis parameters were varied on a linear uniform scale as follows: $N^* = [0.01 \dots 1]$ OD₇₀₀, $r_{i,t}^* = [0.01 \dots 1]$ (unitless ratio), $x_{i,t}^* = [0 \dots 75]$ mg/L, $\gamma_{i,t} = [0.01 \dots 1]$ (biomass yield per g or mg), $V_{+,i,t} = [1 \dots 30]$ g/h, $K_{-,i,t} = [1 \dots 100]$ g, $V_{+,i,t} = [1 \dots 120]$ mg/h, $K_{-,i,t} = [1 \dots 1000]$ mg, $\phi_{i,t} = [0.01 \dots 0.5]$ (unitless ratio).

Nominal parameters. We choose the following nominal parameters: $\gamma_{i,t} = 0.05$, $\gamma_{i,t} = 0.5$, $V_{+,i,t} = 7.2$, $K_{-,i,t} = 5$, $\eta_{i,t} = 0.001$, $V_{+,i,t} = 30$, $K_{-,i,t} = 50$, $\delta_{i,t} = 1$. These

parameters are in the middle of the uptake ranges for the strains in this study (Figure S17). We choose $\phi_{i,t} = 0.1$ based on our analysis in Figure 1. We set the initial concentration of glucose $G(t = 0) = 20$ g per L and exchange metabolite $x_{i,t}(t = 0) = 0$ mg per L (if absent) or 75 mg per L if present. The initial total population is $N(t = 0) = 0.03$ OD₇₀₀. The initial population of each species is $y_{i,t}(t = 0) = r_{i,t}^* \cdot N(t = 0)$ where $r_{i,t}^*$ is the proportion. We simulate all models for a time span of 168 hours.”

Minor comments:

There is some mention of previous work using external control methods for microbial communities, but there is no mention of previous work on the kinds of self-regulating communities generated here. In particular:

10.1038/s41467-020-20756-2 for work on community design

10.1038/s41467-021-22240-x for experimental work, performing many of the same sort of characterisation methods described here.

We thank the reviewer for providing valuable feedback regarding the missing references. We have now incorporated the two suggested references into the revised manuscript.

Line 62 “Progress on establishing cross-feeding *E. coli* communities has been made” – references

We have now added three references for this statement in the revised manuscript.

Line 107 “it is” -> its

Revised “it is” to “its” as suggested.

Line 119-120 “two strains, denoted $i = 1$ (producing metabolite 2) and $i = 2$ (producing metabolite 1)” – Fig 1A shows J1 producing x1 not x2

We have amended the text to reflect Figure 1A and the model formation in the Supplementary Material.

Line 152 “traceable” -> tractable

Revised “traceable” to “tractable” as suggested.

Fig 4G – comment on the drop in the red population during stationary phase

The decline in the red population seems to follow the decrease observed in overall OD700, indicating that the red population may enter an ageing process (DOI: [10.3390/fermentation5020037](https://doi.org/10.3390/fermentation5020037)) – the reduction in absorbance can be attributed to cell morphology changes that can take place at stationary phase (although it can be also lined to sedimentation/condensation effects). In addition, reduction in red fluorescence could be related to the degradation of the RPF during the stationary phase, which could be attributed to either nutrient depletion, limited oxygen availability, or a combination of both factors.

Fig S16E – initial densities were 0.4 but max densities were 0.2. Please comment.

I believe there are multiple measures of OD being reported; OD600 and OD700 from a spectrophotometer and from a plate reader. This is currently making it difficult to compare certain data and follow changes e.g. Fig S16E, initial densities were 0.4 but max densities were 0.2. This should be easy to rectify by calibrating the machines so data can be reported on the same scales. See 10.1101/803239, 10.1021/acssynbio.0c00296.

We appreciate your valuable input regarding the different OD (optical density) measurements and equipment. Following a comparable data processing strategy to the one proposed, in the revised manuscript we recalibrated all OD values to the OD_{700nm} scale using a Tecan plate reader (the calibration curve can be found in Tab S9). In the revised Figure S22E (previous Figure S16E), the recalculated initial density of OD_{700nm} 0.078 is now displayed instead of the original OD_{600nm} 0.4, making the results, as suggested by the reviewer, easier to follow and compare.

Line 491 Greiner 655090 are cell-culture treated plates. Does this have any effect on yeast growth, protein/metabolite binding and availability etc.?

The 96-well microplates with μ Clear® bottom, specifically the Greiner 655090 model were selected for their superior performance in fluorescence readings. They have been functionalised to have a polar surface. While this can help cell-surface interactions in certain mammalian cells when incubated at low-to-none agitation conditions, it should not be the case in yeast, where the culture is shaken vigorously and in which cell-cell or cell-surface interactions are related to hydrophobic interactions (e.g. via flo proteins <https://elifesciences.org/articles/55587>).

In our preliminary experiments (Fig. S11) and subsequent co-culture investigations, we observed that the cell-culture treated surface of these plates had no discernible impact on yeast growth, and therefore nutrient availability. We found reproducible OD results with other microplates such as Corning® 96 well plates (CLS4591), which we use when it is not required to measure any fluorescent signal.

Furthermore, these 96-well plates (Greiner 655090) have been extensively utilized and acknowledged in many articles for yeast cultures, for example:

- <https://doi.org/10.1371/journal.pone.0025136>
- <https://www.nature.com/articles/s41598-019-38913-z>
- <https://doi.org/10.1016/j.ijfoodmicro.2012.03.028>
- <https://www.nature.com/articles/s41467-018-03191-2>
- <https://www.nature.com/articles/s41589-023-01341-2>
- <https://www.pnas.org/doi/abs/10.1073/pnas.1908571116>
- <https://www.sciencedirect.com/science/article/pii/S2589004223019399>
- [https://www.cell.com/iscience/fulltext/S2589-0042\(23\)01939-9](https://www.cell.com/iscience/fulltext/S2589-0042(23)01939-9)

Fig 1 B – it is not clear if this modelling is performed for monocultures or co-cultures.

This analysis is of production and growth rates from a monoculture model. We have amended the caption as follows: “**B.** *Initial simulations of the impact of metabolite production on host growth in yeast monocultures.*”

Fig 1B right – is this output at a particular time?

Figure 1B depicts maximal growth rate, and for clarity the figure legend has been amended to: “*(right) Metabolite production has a nonlinear relationship with maximal growth.*”

Fig 2D – I’m not sure I understand the categorisations of high, medium and low

As mentioned in responses to the reviewer 1 and 2, we have overhauled how we delineate the degree of strain proficiency in promoting co-culture growth (i.e., strong/medium/weak). We have now performed ANOVA tests, followed by Bonferroni’s multiple comparisons tests on the growth of co-cultures, and established an arbitrary thresholds for the definition of strong ($OD_{700nm} \geq 0.5$), medium ($0.3 \leq OD_{700nm} < 0.5$) and weak ($OD_{700nm} < 0.3$) which has consequently reclassified the co-cultures in Figure 2D, main text, and Supplementary Note 5.

“Based on the growth (OD_{700nm}) of the co-cultures, we classified each target metabolite by their ability to facilitate growth in cross-feeding co-cultures: strong ($OD_{700nm} \geq 0.5$): ade, trp, met, his; medium ($0.3 \leq OD_{700nm} < 0.5$): lys, phe&tyr, val&ile, cys, leu, ura; and weak ($OD_{700nm} < 0.3$): thr, tyr, arg, ser.”

Fig 5C – Adding his reduces the his auxotroph proportion? Please comment.

In the absence of exchanged metabolite supplementation, the co-culture AKH_III exhibited the dominance of the ade auxotrophic strain (tagged with RFP), indicating its higher growth compared to the other two auxotrophic strains. The addition of histidine (his) to the co-culture significantly benefited the growth of the his auxotrophic strain (tagged with GFP), consequently leading to increased GFP-tagged populations. Interestingly, the increase in the his auxotroph (GFP-tagged) population was the highest at a his supplementation level of 10 mg/L and then reduced (although still higher than without his addition) at increased his concentrations of 20 mg/L and even more at 40 mg/L. A possible explanation for this is that the increase in GFP-tagged populations contributed to the overproduction of adenine, thereby supporting the accelerated growth of the ade auxotrophic strain (tagged with RFP), making the relative proportion of his auxotrophic strains to seem reduced at 20 or 40 mg/L when compared with 10 mg/L.

Decision Letter, first revision:

Message: Our ref: NMICROBIOL-23041016A

24th November 2023

Dear Rodrigo,

Thank you for your patience as we've prepared the guidelines for final submission of your Nature Microbiology manuscript, "Engineering synthetic yeast communities with a molecular cross-feeding toolkit" (NMICROBIOL-23041016A). Please carefully follow the step-by-step instructions provided in the attached file, and add a response in each row of the table to indicate the changes that you have made. Ensuring that each point is addressed will help to ensure that your revised manuscript can be swiftly handed over to our production team.

If you have not done so already, please alert us to any related manuscripts from your group that are under consideration or in press at other journals, or are being written up for submission to other journals (see: <https://www.nature.com/nature-research/editorial->

policies/plagiarism#policy-on-duplicate-publication for details).

In recognition of the time and expertise our reviewers provide to Nature Microbiology's editorial process, we would like to formally acknowledge their contribution to the external peer review of your manuscript entitled "Engineering synthetic yeast communities with a molecular cross-feeding toolkit". For those reviewers who give their assent, we will be publishing their names alongside the published article.

Nature Microbiology offers a Transparent Peer Review option for new original research manuscripts submitted after December 1st, 2019. As part of this initiative, we encourage our authors to support increased transparency into the peer review process by agreeing to have the reviewer comments, author rebuttal letters, and editorial decision letters published as a Supplementary item. When you submit your final files please clearly state in your cover letter whether or not you would like to participate in this initiative. Please note that failure to state your preference will result in delays in accepting your manuscript for publication.

Cover suggestions

COVER ARTWORK: We welcome submissions of artwork for consideration for our cover. For more information, please see our [a href="https://www.nature.com/documents/Nature_covers_author_guide.pdf" target="new">guide for cover artwork](https://www.nature.com/documents/Nature_covers_author_guide.pdf).

Nature Microbiology has now transitioned to a unified Rights Collection system which will allow our Author Services team to quickly and easily collect the rights and permissions required to publish your work. Approximately 10 days after your paper is formally accepted, you will receive an email in providing you with a link to complete the grant of rights. If your paper is eligible for Open Access, our Author Services team will also be in touch regarding any additional information that may be required to arrange payment for your article.

Please note that *Nature Microbiology* is a Transformative Journal (TJ). Authors may publish their research with us through the traditional subscription access route or make their paper immediately open access through payment of an article-processing charge (APC). Authors will not be required to make a final decision about access to their article until it has been accepted. [a href="https://www.springernature.com/gp/open-research/transformative-journals"> Find out more about Transformative Journals](https://www.springernature.com/gp/open-research/transformative-journals)

Authors may need to take specific actions to achieve [a href="https://www.springernature.com/gp/open-research/funding/policy-compliance-faqs">compliance with funder and institutional open access mandates.](https://www.springernature.com/gp/open-research/funding/policy-compliance-faqs) If your research is supported by a funder that requires immediate open access (e.g. according to [a href="https://www.springernature.com/gp/open-research/plan-s-compliance">Plan S principles\)\) then you should select the gold OA route, and we will direct you to the compliant route where possible. For authors selecting the subscription publication route, the journal's standard licensing terms will need to be accepted, including \[a href="https://www.nature.com/nature-portfolio/editorial-policies/self-archiving-and-license-to-publish">self-archiving policies. Those licensing terms will supersede any other terms that the author or any third party may assert apply to any version of the manuscript.\]\(https://www.nature.com/nature-portfolio/editorial-policies/self-archiving-and-license-to-publish\)](https://www.springernature.com/gp/open-research/plan-s-compliance)

Best regards,

Reviewer #1:
Remarks to the Author:
No further comments.

Reviewer #2:
Remarks to the Author:
The authors have made extensive revisions to address my comments and substantially improved the manuscript. However, there are a few minor points to address.

1. Line 190-191: Mutant alleles should be written in small letters such as $ade8\Delta$. Please check with the journal's guidelines and follow the standard yeast genetics nomenclature.

2. The research offers a valuable toolkit for creating synthetic consortia. Yet, it may be challenging for other researchers to determine the most promising combination to test from the numerous options available. It would be beneficial if the discussion section could suggest some practical uses for the toolkit and advise on which combinations might be most effective for specific applications.

Reviewer #3:
Remarks to the Author:
Thank you to the authors for taking on board my comments and the comments of the other reviewers. I am satisfied with the responses and the extra work carried out.

Author Rebuttal, first revision:

We would like to thank you and the reviewers for your constructive feedback on our revised manuscript. Please find below our point-by-point responses to the comments made by Reviewer #2. Additionally, in compliance with the provided author checklist, we have restructured the manuscript and updated all sections of the main text and supplemental information to align with the journal's guidelines. We are confident that our manuscript now adheres to these guidelines, and we believe it is now suitable for publication.

Besides, twitter accounts for the first author and corresponding authors are @HuadongPeng and @LedesmaAmaro, respectively.

Sincerely,

Rodrigo Ledesma-Amaro

Reviewer #2:

Remarks to the Author:

The authors have made extensive revisions to address my comments and substantially improved the manuscript. However, there are a few minor points to address.

1. Line 190-191: Mutant alleles should be written in small letters such as $ade8\Delta$. Please check with the journal's guidelines and follow the standard yeast genetics nomenclature.

We appreciate the comment. We have double checked and corrected the nomenclature between the wild type and mutant forms of the yeast genes in the whole manuscript, including supplementary figures and materials.

2. The research offers a valuable toolkit for creating synthetic consortia. Yet, it may be challenging for other researchers to determine the most promising combination to test from the numerous options available. It would be beneficial if the discussion section could suggest some practical uses for the toolkit and advise on which combinations might be most effective for specific applications.

We appreciate the comment. Following your suggestion, we have updated the discussion section to help provide a flavor (kept short due to word limitations) on the selection of co-culture combinations for different applications. The new sentence reads as follows:

“Different co-cultures show distinct features (e.g. different growth rates, population dynamics, final biomass), which can be used to guide their selection based on the desired application (e.g. mimicking behaviours observed on wild communities, or balancing biomass production to maximise product formation.”

Final Decision Letter:

Message: 18th December 2023

Dear Rodrigo,

I am pleased to accept your Resource "A molecular toolkit of cross-feeding strains for engineering synthetic yeast communities" for publication in Nature Microbiology. Thank you for having chosen to submit your work to us and many congratulations.

You may wish to make your media relations office aware of your accepted publication, in case they consider it appropriate to organize some internal or external publicity. Once your paper has been scheduled you will receive an email confirming the publication details. This is normally 3-4 working days in advance of publication. If you need additional notice of the date and time of publication, please let the production team know when you receive the proof of your article to ensure there is sufficient time to coordinate. Further information on our embargo policies can be found here:

<https://www.nature.com/authors/policies/embargo.html>

Please note that *Nature Microbiology* is a Transformative Journal (TJ). Authors may publish their research with us through the traditional subscription access route or make their paper immediately open access through payment of an article-processing charge (APC). Authors will not be required to make a final decision about access to their article until it has been accepted. [Find out more about Transformative Journals](https://www.springernature.com/gp/open-research/transformative-journals)

Authors may need to take specific actions to achieve [compliance](https://www.springernature.com/gp/open-research/funding/policy-compliance-faqs) with funder and institutional open access mandates. If your research is supported by a funder that requires immediate open access (e.g. according to [Plan S principles](https://www.springernature.com/gp/open-research/plan-s-compliance)) then you should select the gold OA route, and we will

nature portfolio

direct you to the compliant route where possible. For authors selecting the subscription publication route, the journal's standard licensing terms will need to be accepted, including [self-archiving policies](https://www.nature.com/nature-portfolio/editorial-policies/self-archiving-and-license-to-publish). Those licensing terms will supersede any other terms that the author or any third party may assert apply to any version of the manuscript.

Congratulations once again and I look forward to seeing the article published.

With kind regards,